# Marginal-Nonuniform PAC Learnability

**Steve Hanneke**
Purdue University
steve.hanneke@gmail.com

**Shay Moran**
Technion and Google Research
smoran@technion.ac.il

**Maximilian Thiessen**
TU Wien
maximilian.thiessen@tuwien.ac.at

## Abstract

We revisit the classical model of nonuniform PAC learning, introduced by Benedek and Itai [1994], where generalization guarantees may depend on the target concept (but not on the marginal distribution). In this work, we study a complementary variant, which we call marginal-nonuniform learning. In this setting, guarantees may depend on the marginal distribution over the domain, but must hold uniformly over all concepts. This captures the intuition that some data distributions are inherently easier to learn from than others, allowing for a flexible, distribution-sensitive view of learnability. Our main result is a complete characterization of the achievable learning rates in this model, revealing a trichotomy: exponential rates of the form $e^{-n}$ arise precisely when the hypothesis class is finite; linear rates of the form $d/n$ are achievable when a recently introduced combinatorial parameter, the VC-eluder dimension $d$, is finite; and arbitrarily slow rates may occur when $d = \infty$. Additionally, in the original (concept-)nonuniform model, we show that for all learnable classes linear rates are achievable. We conclude by situating marginal-nonuniform learning within the landscape of universal learning, and by discussing its relationship to other distribution-dependent learning paradigms.

## 1 Introduction

We study the possible learning rates for binary classification in a variant of *nonuniform learning* introduced by Benedek and Itai [1994]. In standard PAC learning, generalization guarantees take the form of a *uniform* rate: the learning rate applies simultaneously to all target concepts and all marginal distributions over the data domain. In contrast, the *concept-nonuniform*[1] model of Benedek and Itai [1994] allows the rate to depend on the target concept $f^*$, while still requiring it to hold uniformly over all marginal distributions.

In this work, we study the complementary variant: *marginal-nonuniform learning*, in which the learning rate may depend on the marginal distribution $P$ over the domain, but must hold uniformly over all target concepts once the marginal is fixed. This model captures the intuition that some data distributions may be inherently easier to learn from than others, while still requiring a single algorithm to generalize well for all possible labelings under a fixed marginal.

Together, these choices give rise to four basic notions of PAC learnability, depending on whether the learning rate is uniform or nonuniform over concepts and distributions. This taxonomy was introduced by Ben-David, Benedek, and Mansour [1995], who studied learnability in all four cases.

---

[1] In the literature this is simply known as nonuniform learning. We use the term concept-nonuniform to distinguish it from marginal-nonuniform learning studied here.

39th Conference on Neural Information Processing Systems (NeurIPS 2025).

In the marginal-nonuniform setting, they exhibited examples showing that different rates can arise depending on the hypothesis class.

In this work, we go further and provide a complete characterization of the achievable rates in the marginal-nonuniform regime. Our main result reveals a clean trichotomy: exponential rates of the form $e^{-n}$ arise precisely when the hypothesis class is finite; linear rates of the form $d/n$ are achievable when a recently introduced combinatorial parameter—the *VC-eluder dimension* $d$—is finite; and arbitrarily slow rates may occur when $d = \infty$.

(Concept-)nonuniform learning is well-known [Benedek and Itai, 1994, Vapnik, 1998, Lugosi and Zeger, 1996, Shalev-Shwartz and Ben-David, 2014] and inspired various practical learning paradigms, such as, structural risk minimization [Vapnik, 1998], Occam's razor [Blumer, Ehrenfeucht, Haussler, and Warmuth, 1987], and the minimum description length principle [Rissanen, 1978]. Concept-nonuniform learning allows to consider a broader family of *learnable* classes than uniform learning, where the latter is characterized by the finiteness of the VC dimension. Moreover, nonuniformity allows to distinguish easy to learn concepts from hard to learn ones and through that potentially achieve better learning rates with smaller constants for specific concepts, which would not be possible with worst-case uniform rates. Similarly, marginal-nonuniform learning allows to distinguish between easy to learn marginal distributions and hard to learn ones. For example, linear classifiers are learnable with lower sample complexity for distributions supported on a subspace of smaller dimension.

**Further related work.**  Marginal-nonuniform learning is related to learning with a single fixed (and potentially even known) distribution, see Benedek and Itai [1991] and to recent results on learnability for (families of) fixed distributions [Lechner and Ben-David, 2024, Hopkins, Kane, Lovett, and Mahajan, 2025]. Moreover, our work continues the line of work on universal learning and their learning rates [Bousquet, Hanneke, Moran, van Handel, and Yehudayoff, 2021, Blanchard, 2022, Hanneke, Karbasi, Moran, and Velegkas, 2024, Attias, Hanneke, Kalavasis, Karbasi, and Velegkas, 2024]. Recently, Hanneke and Xu [2024] studied the possible universal rates for (worst-case) empirical risk minimizers (ERMs) in contrast to the optimal learners proposed by Bousquet et al. [2021]. Interestingly, one of the parameters introduced by Hanneke and Xu [2024]—the *VC-eluder dimension*—is also characterizing learnability in our marginal-nonuniform setting. Indeed, one of our main results is that the VC-eluder dimension $d = \mathrm{VCE}(\mathcal{H})$ characterizes the exact *fine-grained* rate $d/n$ of marginal-nonuniform learning. Here, fine-grained learning rates were introduced by Bousquet, Hanneke, Moran, Shafer, and Tolstikhin [2023] allowing to state parameter dependent rates like $\mathrm{VC}(\mathcal{H})/n$, instead of simply $1/n$, with sharper control over the tails of the learning curves. For a discussion on the relationship of the VC-eluder dimension with relevant parameters from universal learning, see Hanneke and Xu [2024]. Separations between distribution independent and distribution dependent learning, as we study here, were considered before by Feldman [2017] in the statistical query setting.

**Overview of main contributions.**

1. We initiate the study of marginal-nonuniform learning rates and provide a complete characterization of the rates that can arise in this setting, depending on natural parameters of the hypothesis class. Our main result is a trichotomy of possible rates (Theorem 3) along with a fine-grained characterization of the linear regime (Theorem 8).

2. Our characterization reveals that the family of hypothesis classes that are marginal-nonuniformly learnable with a linear rate coincides with the family that are uniformly learnable with a linear rate—that is, the PAC learnable classes. At first glance, this may seem to suggest that allowing the constants to depend on the marginal offers no additional power. However, a closer look shows that the linear rate in marginal-nonuniform learning depends on the $\mathrm{VCE}(\mathcal{H})$ dimension, whereas the rate in uniform PAC learning depends on the $\mathrm{VC}(\mathcal{H})$ dimension. Since $\mathrm{VCE}(\mathcal{H}) \leq \mathrm{VC}(\mathcal{H})$ always, and the gap can be arbitrarily large (e.g., when $\mathrm{VCE}(\mathcal{H}) = 1$ but $\mathrm{VC}(\mathcal{H})$ is unbounded), this distinction is significant.

3. We revisit known results on concept-nonuniform learnability and provide an improvement over the previously established $\log n/n$ rate [Lugosi and Zeger, 1996], showing that a linear rate of $1/n$ is achievable and optimal in this setting.

4. We discuss the relationships between the possible learning rates across all four learning settings; see Figure 1 and Section 5. We also highlight open problems from both the existing literature and our own work (Section 6).

## 2 Preliminaries and main results

Let $X$ be a domain and $\mathcal{H} \subseteq \{0,1\}^X$ a hypothesis space. We denote by $\Delta(X)$ the set of all distributions on $X$ and call them marginal distributions (in contrast to distributions on $X \times \{0,1\}$). For every $P \in \Delta(X)$ and $f^*, h \in \mathcal{H}$ let $\mathrm{er}_{P,f^*}(h) = \mathrm{Pr}_{x \sim P}(f^*(x) \neq h(x))$ be the expected error of hypothesis $h$ under the distribution $P$ with points labeled by $f^*$. This is a strict version of the *realizability* assumption, see Section 6 for a discussion. Fix $P, f^*$. Let $\hat{h}_n = A(S_n)$ be the output hypothesis of a learning algorithm $A$ taking a sample $S_n = ((x_i, y_i))_{i=1}^n$ of size $n$ with $x_i$ iid from $P$ and $y_i = f^*(x_i)$. We write $\mathbb{E}[\cdot] = \mathbb{E}_{S_n}[\cdot]$ as the expectation over such a sample $S_n$. We denote by $S_{\leq \ell}$ the subsequence of length $\ell \in \mathbb{N}$ of $S$.

The *version space induced by* $S_n$ is $\mathrm{VS}(S_n, \mathcal{H}) = \{h \in \mathcal{H} \mid h(x_i) = y_i \text{ for all } i \in [n]\}$. A set $S \subseteq X$ is shattered if for all $y : S \to \{0,1\}$ there exists $h \in \mathcal{H}$ such that $h(x) = y(x)$ for all $x \in S$. The VC dimension $\mathrm{VC}(\mathcal{H})$ of $\mathcal{H}$ is the size of a largest shattered set. If there exist shattered sets of all sizes $d \in \mathbb{N}$ then $\mathrm{VC}(\mathcal{H}) = \infty$. We call $\mathcal{H}$ a *VC-class* if the VC dimension $\mathrm{VC}(\mathcal{H})$ of $\mathcal{H}$ is finite. We will use the following related, recently introduced combinatorial parameter.

**Definition 1** (VC-eluder dimension, Hanneke and Xu 2024). *Let $\mathcal{H}$ be a hypothesis space, $d \in \mathbb{N}$, and $h \in \mathcal{H}$. We say $\mathcal{H}$ has an infinite $d$-VC-eluder sequence $S = \{(x_1, h(y_1)), (x_2, h(y_2)), \dots\}$ centered at $h$ if $\{x_{kd+1}, \dots, x_{kd+d}\}$ is shattered by $\mathrm{VS}(S_{\leq kd}, \mathcal{H})$ for all $k \in \mathbb{N}$. A 1-VC-eluder sequence is an* infinite eluder sequence*. The* VC-eluder dimension *of $\mathcal{H}$ is the largest integer $\mathrm{VCE}(\mathcal{H}) = d \geq 0$ such that $\mathcal{H}$ has an infinite $d$-VC-eluder sequence centered at some $h \in \mathcal{H}$. If $\mathcal{H}$ has an infinite $d$-VC-eluder sequence for all $d \in \mathbb{N}$ then $\mathrm{VCE}(\mathcal{H}) = \infty$.*

By definition $\mathrm{VCE}(\mathcal{H}) \leq \mathrm{VC}(\mathcal{H})$. Moreover, there are families of classes such that $\mathrm{VCE}(\mathcal{H}) = 1$ while $\mathrm{VC}(\mathcal{H})$ is arbitrarily large [Hanneke and Xu, 2024].

A *rate function* is a function $R : \mathbb{N} \to [0,1]$ with $\lim_{n \to \infty} R(n) = 0$. Adapting Bousquet et al. [2021] we have the following cases of learning with rate $R$, which only differ in the order of quantifiers.

**Uniform learning**:

$$\exists \hat{h}_n \text{ s.t. } \exists C, c > 0 \text{ s.t. } \forall P \in \Delta(X) \,\forall f^* \in \mathcal{H}: \mathbb{E}[\mathrm{er}_{P,f^*}(\hat{h}_n)] \leq CR(cn) \text{ for all } n.$$

**Universal learning**:[2]

$$\exists \hat{h}_n \text{ s.t. } \forall P \in \Delta(X), \forall f^* \in \mathcal{H}, \exists C, c > 0: \mathbb{E}[\mathrm{er}_{P,f^*}(\hat{h}_n)] \leq CR(cn) \text{ for all } n.$$

**Concept-nonuniform learning**:

$$\exists \hat{h}_n \text{ s.t. } \forall f^* \in \mathcal{H} \,\exists C, c > 0 \text{ s.t. } \forall P \in \Delta(X): \mathbb{E}[\mathrm{er}_{P,f^*}(\hat{h}_n)] \leq CR(cn) \text{ for all } n.$$

**Marginal-nonuniform learning**:

$$\exists \hat{h}_n \text{ s.t. } \forall P \in \Delta(X) \,\exists C, c > 0 \text{ s.t. } \forall f^* \in \mathcal{H}: \mathbb{E}[\mathrm{er}_{P,f^*}(\hat{h}_n)] \leq CR(cn) \text{ for all } n.$$

More explicitly we define marginal-nonuniform learning with a rate function as follows.

**Definition 2** (Marginal-nonuniform learning). *Let $\mathcal{H}$ be a class and $R$ a rate function.*

1. *$\mathcal{H}$ is marginal-nonuniformly learnable at rate $R$ if there exists a learning algorithm $\hat{h}_n$ such that for all marginal distributions $P \in \Delta(X)$ there exists $C, c > 0$ such that for all target functions $f^* \in \mathcal{H}$ we have $\mathbb{E}[\mathrm{er}_{P,f^*}(\hat{h}_n)] \leq CR(cn)$ for all $n$.*

2. *$\mathcal{H}$ is not marginal-nonuniformly learnable at rate faster than $R$ if for every learning algorithm $\hat{h}_n$ there exists a marginal distributions $P \in \Delta(X)$, constants $C, c > 0$, such that for infinitely many $n$ there exists a target concept $f^* \in \mathcal{H}$ (dependent on $n$) such that $\mathbb{E}[\mathrm{er}_{P,f^*}(\hat{h}_n)] \geq CR(cn)$.*

---

[2]We emphasize that this slightly differs from the standard definition of universal learning [Bousquet et al., 2021] in which a broader set of "*realizable*" distributions is used. See Section 6 for a discussion.

3. $\mathcal{H}$ is marginal-nonuniformly learnable at exact rate $R$ if $\mathcal{H}$ is marginal-nonuniformly learnable at rate $R$ and not faster than rate $R$.

4. $\mathcal{H}$ requires at least arbitrarily slow rates to be marginal-nonuniformly learnable *if for every rate function $R'$, the class $\mathcal{H}$ is not marginal-nonuniformly learnable at rate faster than $R'$.*

5. $\mathcal{H}$ is marginal-nonuniformly learnable (independently of rates) *if there exists a learning algorithm $\hat{h}_n$ and a function $R^* : \mathbb{N} \times \Delta(X) \to [0,1]$ with $\lim_{n\to\infty} R^*(n, P) = 0$ for all $P \in \Delta(X)$ such that for all target concepts $f^* \in \mathcal{H}$ we have $\mathbb{E}[\mathrm{er}_{P,f^*}(\hat{h}_n)] \leq R^*(n, P)$.*

We can now state our main result, a trichotomy of rates for marginal-nonuniform learning.

**Theorem 3** (Marginal-nonuniform trichotomy)**.** *Every class $\mathcal{H}$ with $|\mathcal{H}| \geq 3$ satisfies:*

1. *$\mathcal{H}$ is marginal-nonuniformly learnable with exact rate $e^{-n}$ if and only if $\mathrm{VCE}(\mathcal{H}) = 0$ (equivalently $|\mathcal{H}| < \infty$).*

2. *$\mathcal{H}$ is marginal-nonuniformly learnable with exact rate $1/n$ if and only if $1 \leq \mathrm{VCE}(\mathcal{H}) < \infty$ (equivalently $|\mathcal{H}| = \infty$ and $\mathrm{VC}(\mathcal{H}) < \infty$).*

3. *$\mathcal{H}$ requires at least arbitrarily slow rates to be marginal-nonuniformly learnable if and only if $\mathrm{VCE}(\mathcal{H}) = \infty$ (equivalently $\mathrm{VC}(\mathcal{H}) = \infty$).*

In Theorem 8 we deepen the result for the linear case: we show that the exact fine-grained rate is $\mathrm{VCE}(\mathcal{H})/n$. See Section 3.3 for more details.

# 3   Marginal-nonuniform learning

In this section we give the details of the marginal-nonuniform learning trichotomy by going through the individual cases. Missing proofs can be found in Appendix A.

## 3.1   Exponential rates for finite classes

We first show that all finite classes are marginal-nonuniformly learnable with exact exponential rate.

**Lemma 4.** *Any class $\mathcal{H}$ with $3 \leq |\mathcal{H}| < \infty$ is marginal-nonuniformly learnable with exact rate $e^{-n}$.*

The $|\mathcal{H}| \geq 3$ assumption is simply for excluding trivially learnable classes [Bousquet et al., 2021]. By Lemma 8 of Hanneke and Xu [2024] we know that $\mathcal{H}$ is finite if and only if $\mathrm{VCE}(\mathcal{H}) = 0$ (i.e., $\mathcal{H}$ has no infinite eluder sequence). Hence the if-direction of the first bullet point of Theorem 3 follows.

## 3.2   Linear rates through finite VC dimension

We first show that a class with VC-eluder dimension $1 \leq \mathrm{VCE}(\mathcal{H}) < \infty$ can be marginal-nonuniformly learned with linear rate. Moreover, $\mathrm{VCE}(\mathcal{H})$ is finite if and only if $\mathrm{VC}(\mathcal{H})$ is finite [Hanneke and Xu, 2024, Remark 19]. Thus it suffices to prove the following lemma.

**Lemma 5.** *Every class $\mathcal{H}$ with finite VC-dimension is marginal-nonuniformly learnable at rate $1/n$.*

Next we show that this is the best possible rate.

**Lemma 6.** *Every infinite class $\mathcal{H}$ is not marginal-nonuniformly learnable at rate faster than $1/n$.*

This proves the if-direction of the second bullet point of Theorem 3. However, as we will see in the next section a much stronger bound is possible; while still resulting in a linear rate it is determined by the VC-eluder dimension instead of the VC dimension of the class.

## 3.3   Fine-grained linear rates through finite VC-eluder dimension

Using the notion of fine-grained rates [Bousquet et al., 2023], we deepen the results for the linear case. In particular, we provide fine-grained rates characterized by the VC-eluder dimension. Let us first define fine-grained rates for marginal-nonuniform learning.

**Definition 7** (Marginal-nonuniform fine-grained rates). *Let $\mathcal{H} \subseteq \{0,1\}^X$ be a class and $R$ be a distribution-independent rate function.*

1. *The class $\mathcal{H}$ is* marginal-nonuniformly learnable at fine-grained rate $R$ *if there exists an algorithm $\hat{h}_n$ such that for every distribution $P \in \Delta(X)$ there exists a distribution-dependent rate function $\lambda(n) = o(R(n))$ such that for all $f^* \in \mathcal{H}$ it holds $\mathbb{E}[\mathrm{er}_{P,f^*}(\hat{h}_n)] \leq R(n) + \lambda(n)$.*

2. *The class is* not marginal-nonuniformly learnable at fine-grained rate faster than $R$ *if for each learner $\hat{h}_n$ there exists a distribution $P \in \Delta(X)$ such that for infinitely many $n$ there exists a target function $f^* \in \mathcal{H}$ (dependent on $n$) such that $\mathbb{E}[\mathrm{er}_{P,f^*}(\hat{h}_n)] \geq R(n)$.*

3. *The class is* marginal-nonuniformly learnable at exact fine-grained rate $R$ *if 1. and 2. hold.*

We now state the fine-grained version of the main result of this work.

**Theorem 8** (Fine-grained marginal-nonuniform rate). *Let $\mathcal{H}$ be a class with VC-eluder dimension $\mathrm{VCE}(\mathcal{H}) = d < \infty$. The class $\mathcal{H}$ is marginal-nonuniformly learnable at exact fine-grained rate $d/n$.*

*Proof.* The proof follows from Lemma 9 and Lemma 12. $\qquad\square$

**Lemma 9.** *There exists an absolute constant $\alpha$ such that for each class $\mathcal{H}$ with $\mathrm{VCE}(\mathcal{H}) < \infty$, there exists a learner $\hat{h}_n$ such that for all $f^* \in \mathcal{H}$ and all marginal distributions $P \in \Delta(X)$ we have:*

$$\mathbb{E}[\mathrm{er}_{P,f^*}(\hat{h}_n)] \leq \alpha \frac{\mathrm{VCE}(\mathcal{H})}{n} + e^{-\kappa(P)n},$$

*where $\kappa(P)$ is a marginal-distribution-dependent (but target concept independent) constant.*

*Proof.* Let $P$ be a marginal distribution, $f^*$ be the target concept, let $m = n/2$, and $S_m, S'_m \in (X \times Y)^m$ be two iid samples of size $m$ from $P$ labeled by $f^*$. Denote by $A$ the event that the version space $\mathrm{VS}(S_m, \mathcal{H})$ has VC dimension at most $d = \mathrm{VCE}(\mathcal{H})$ on $S'_m$. Let $\hat{h}_{\mathrm{OIG}}$ be the classifier corresponding to the one-inclusion graph algorithm on the sample $S'_m$ with hypothesis class $\mathcal{H}' = \mathrm{VS}(S_m, \mathcal{H})$. By the law of total expectation (and as the error is bounded by one) we have

$$\mathbb{E}[\mathrm{er}_{P,f^*}(\hat{h}_{\mathrm{OIG}})] \leq \Pr(\neg A) + \mathbb{E}[\mathrm{er}_{P,f^*}(\hat{h}_{\mathrm{OIG}}) \mid A].$$

By Lemma 11 there exists $\kappa' = \kappa'(P)$ such that $\Pr(\neg A) \leq e^{-\kappa' m} = e^{-\frac{\kappa'}{2} n}$. If $A$ holds the version space has VC dimension at most $d$ on $S'_m$. Under this event, we bound the error of $\hat{h}_{\mathrm{OIG}}$ by $\alpha' \frac{d}{m} = 2\alpha' \frac{d}{n}$ for some absolute constant $\alpha'$ [Haussler, Littlestone, and Warmuth, 1994]. $\qquad\square$

For the upper bound we rely on an application of *König's lemma*, which characterizes the finiteness of a tree by the finiteness of all its subpaths.

**Lemma 10** (König's lemma, König 1927). *Let $T$ be a rooted tree with a finite or countable number of nodes $V$ such that each node has a finite number of neighbours. Then $|V|$ is finite if and only if every root to leaf path has finite length.*

The next lemma is the core result allowing marginal-nonuniform learning with rate depending on $\mathrm{VCE}(\mathcal{H})$ instead of $\mathrm{VC}(\mathcal{H})$.

**Lemma 11.** *Let $\mathcal{H}$ be a class with VC-eluder dimension $\mathrm{VCE}(\mathcal{H}) = d < \infty$ and $P \in \Delta(X)$ a marginal distribution. There exists a constant $C$ only depending on $P$ such that with probability at least $1 - e^{-Cn}$ over a sample $S_n \sim P^n$ labeled by an arbitrary $f^* \in \mathcal{H}$, the version space $\mathrm{VS}(S_{\leq n/2}, \mathcal{H})$ induced by the first half of the sample $S_{\leq n/2}$ has, with probability one, VC dimension at most $d$ on the second half of the sample $S \setminus S_{\leq n/2}$.*

*Proof.* Let $d = \mathrm{VCE}(\mathcal{H})$. Fix a marginal distribution $P$ over $X$. Consider an infinite sequence $\bar{S} = \{x_1, x_2, \dots\}$ of iid samples from $P$. Let $T$ be a rooted binary tree given by all possible realizable classifications over $\bar{S}$. More precisely, the nodes in each level $i$ of $T$ correspond to $x_i$, and the two descending edges of each node correspond to labeling the respective $x_i$ either as positive or negative. A descending edge is only added to a node if the path from the root to this node including

the label given by the edge is realizable (i.e., there exists a hypothesis $h \in \mathcal{H}$, such that $h(x_i) = y_i$ with $y_i$ given by the edges). In other words, $T$ is a Littlestone tree, which is level-constrained to the sequence $\bar{S}$ [Littlestone, 1988]. With every node in $T$ we associate a version space given by all hypotheses that realize the classifications on the path leading from the root to this node. Finally, we truncate $T$ at a node whenever its associated version space has VC dimension at most $d$ on the remaining sequence.

**Claim.** *Every path starting in the root of $T$ has finite length.*

Assume there is an infinite path $p$ starting in the root of $T$. Take the first node on $p$. The version space associated with this node has VC dimension at least $d + 1$ on the remaining sequence. Let $S_1$ be a shattered set of size $d + 1$ on the remaining sequence. As the tree is level-constrained, we can reach all nodes corresponding to $S_1$ on $p$ after a finite number of steps. The node after $S_1$ on $p$ again has an associated version space with VC dimension at least $d + 1$ on the remaining sequence. We can thus repeat this argument to get an infinite sequence of version spaces and corresponding shattered sets $S_1, S_2, \ldots$ leading to a $(d + 1)$-VC-eluder sequence, which is not possible. This proves the claim.

Thus, by König's lemma, the tree $T$ is finite and in particular has finite depth. Denote by $\mathrm{depth}(S)$ the integer random variable indicating the depth of $T$ induced by $S$. As $\mathrm{depth}(S)$ is always finite, the median $q_P$, i.e., $q_P = \min\{n \in \mathbb{N} \mid \Pr_{S \sim P^{\mathbb{N}}}(\mathrm{depth}(S) \leq n) \geq 1/2\}$, is also a finite constant (only dependent on $P$). That is, the version space $\mathrm{VS}(S_{\leq q_P}, \mathcal{H})$ of a sample $S_{\leq q_P}$ of size $q_P$ (labeled by any target) has, with probability at least $1/2$, a VC dimension of at most $d$ on the remaining sequence.

This has further consequences. Specifically, for any target $f^* \in \mathcal{H}$, denote by $\tilde{p} = \Pr_{S' \sim P^{d+1}}(\mathrm{VS}(S_{\leq q_P}, \mathcal{H})$ shatters $S')$. We claim that with probability at least $1/2$ (over $S_{\leq q_P}$) we have $\tilde{p} = 0$. To see this, note that since $S_{\leq q_P}$ may be regarded as the first $q_P$ examples from an infinite sequence $S \sim P^{\mathbb{N}}$, and is independent of the remainder of the sequence, given $\tilde{p}$ the conditional probability that $\mathrm{VS}(S_{\leq q_P}, \mathcal{H})$ has VC dimension at most $d$ on the remaining sequence (i.e., does not shatter any $d + 1$ examples in the remaining sequence) is at most $\lim_{r \to \infty}(1 - \tilde{p})^r$, which is $0$ unless $\tilde{p} = 0$. Thus, with probability one, if $\tilde{p} > 0$ then $\mathrm{VS}(S_{\leq q_P}, \mathcal{H})$ does not have VC dimension at most $d$ on the remaining sequence. Since the latter occurs with probability at most $1/2$, we conclude that with probability at least $1/2$ we have $\tilde{p} = 0$.

Now, for any $\ell \in \mathbb{N}$, if we have $\ell$ independent samples of size $q_P$, all labeled by a common target concept, since the version space $\mathrm{VS}(S_{\leq \ell \cdot q_P}, \mathcal{H})$ for the total set of $\ell \cdot q_P$ examples is the intersection of the $\ell$ version spaces for the $\ell$ samples of size $q_P$, the above implies that with probability at least $1 - (\frac{1}{2})^\ell$, we have $\Pr_{S' \sim P^{d+1}}(\mathrm{VS}(S_{\leq \ell \cdot q_P}, \mathcal{H})$ shatters $S') = 0$. In particular, given this occurs, the conditional probability that $\mathrm{VS}(S_{\leq \ell q_P}, \mathcal{H})$ has VC dimension at most $d$ on an additional independent $n/2$ examples is $1$.

Overall, if we draw a sample of size $n$, we have that the version space induced by the first $n/2$ samples has VC dimension at most $d$ on the other $n/2$ samples with probability at least $1 - (\frac{1}{2})^{\lfloor \frac{n/2}{q_P} \rfloor}$. This finishes the proof. $\qquad\square$

**Lemma 12.** *There exists an absolute constant $\beta$ such that for each class $\mathcal{H}$ with $\mathrm{VCE}(\mathcal{H}) < \infty$, for all learners $\hat{h}_n$ there is a marginal distribution $P \in \Delta(X)$, such that for infinitely many $n \in \mathbb{N}$ there exists a target concept $f_n^*$ with*

$$\mathbb{E}[\mathrm{er}_{P, f_n^*}(\hat{h}_n)] \geq \beta \frac{\mathrm{VCE}(\mathcal{H})}{n} \, .$$

### 3.4 Arbitrarily slow rates

The third case of Theorem 3 is given by the following lemma.

**Lemma 13.** *Let $\mathcal{H}$ be a class with infinite VC-eluder dimension. Marginal-nonuniformly learning $\mathcal{H}$ requires at least arbitrarily slow rates.*

Overall, as these three cases, distinguished by $\mathrm{VCE}(\mathcal{H})$, are disjoint and cover all possibilities, this completes the proof of Theorem 3.

# 4 Concept-nonuniform learning

In this section, we recap known results on concept-nonuniform learnability and state an improvement over the best rate known in the literature [Lugosi and Zeger, 1996]. Missing proofs are in Appendix B. We start again with a definition of learning with a specific rate.

**Definition 14** (Concept-nonuniform learning). *Let $\mathcal{H}$ be a class and $R$ a rate function.*

1. *$\mathcal{H}$ is concept-nonuniformly learnable at rate $R$ if there exists a learning algorithm $\hat{h}_n$ such that for all target functions $f^* \in \mathcal{H}$ there exists $C, c > 0$ such that for all marginal distributions $P \in \Delta(X)$ we have $\mathbb{E}[\mathrm{er}_{P,f^*}(\hat{h}_n)] \leq CR(cn)$ for all $n$.*

2. *$\mathcal{H}$ is not concept-nonuniformly learnable at rate faster than $R$ if for every learning algorithm $\hat{h}_n$ there exists a target function $f^* \in \mathcal{H}$ and constants $C, c > 0$, such that for infinitely many $n$ there exists a marginal distribution $P \in \Delta(X)$ (dependent on $n$) such that $\mathbb{E}[\mathrm{er}_{P,f^*}(\hat{h}_n)] \geq CR(cn)$.*

3. *$\mathcal{H}$ is concept-nonuniformly learnable at exact rate $R$ if $\mathcal{H}$ is concept-nonuniformly learnable at rate $R$ and not faster than rate $R$.*

4. *$\mathcal{H}$ is concept-nonuniformly learnable (independently of rates) if there exists a learning algorithm $\hat{h}_n$ and a function $R^* : \mathbb{N} \times \mathcal{H} \to [0,1]$ with $\lim_{n\to\infty} R^*(n,h) = 0$ for all $f^* \in \mathcal{H}$ such that for all marginal distributions $P \in \Delta(X)$ we have $\mathbb{E}[\mathrm{er}_{P,f^*}(\hat{h}_n)] \leq R(n, f^*)$.*

The following characterization of concept-nonuniform learnability is well known.

**Lemma 15** (Benedek and Itai 1994). *A class $\mathcal{H}$ is concept-nonuniformly learnable (independently of rates) if and only if $\mathcal{H}$ is a countable union of VC classes.*

Without loss of generality we can assume that a concept-nonuniformly learnable class $\mathcal{H}$ is a union $\mathcal{H} = \bigcup_{i=1}^{\infty} \mathcal{H}_i$ of a nested sequence $\mathcal{H}_1 \subseteq \mathcal{H}_2 \subseteq \ldots$ of VC classes. Common approaches rely on empirical risk minimization (ERM) on each VC class $\mathcal{H}_i$, which leads to an overall $\log n / n$ rate [Lugosi and Zeger, 1996]. By using a combination of one-inclusion graph predictors [Haussler et al., 1994] with an online-to-batch conversion instead, we achieve the optimal linear rate $1/n$. This leads to the following dichotomy.

**Theorem 16** (Concept-nonuniform dichotomy). *For every class $\mathcal{H}$ with $|\mathcal{H}| \geq 3$ the following holds:*

1. *$\mathcal{H}$ is concept-nonuniformly learnable with exact rate $1/n$ if $\mathcal{H}$ is a countable union of VC classes.*

2. *$\mathcal{H}$ is not concept-nonuniformly learnable if $\mathcal{H}$ is not a countable union of VC classes.*

The proof of Theorem 16 follows from the following two lemmas.

**Lemma 17.** *Let $\mathcal{H}$ be a countable union of VC classes. The class $\mathcal{H}$ is concept-nonuniformly learnable at rate $1/n$.*

*Proof.* If $\mathcal{H}$ has finite VC dimension we just run the one-inclusion graph algorithm and obtain a linear error rate [Haussler et al., 1994]. Thus we can assume $\mathrm{VC}(\mathcal{H}) = \infty$. Without loss of generality $\mathcal{H}$ is a nested union $\mathcal{H}_1 \subseteq \mathcal{H}_2 \subseteq \ldots$ of VC classes (we can set $\mathcal{H}'_i = \bigcup_{j=1}^{i} \mathcal{H}_j$). Denote by $d_1 \leq d_2 \leq \ldots$ the respective VC dimensions. Since $d_i \to \infty$, we can moreover assume that $d_i \geq \ln(i+1)$ for all $i$, again by taking unions of subclasses. Let $S$ be an iid sample of size $n$ from a distribution $P$ over $X$ labeled by some $f^* \in \mathcal{H}$ and define $i^* = \min\{i \mid f^* \in \mathcal{H}_i\}$. We use the first half $S'$ of $S$ to train a classifier for each $\mathcal{H}_i$ and the second half $S''$ of $S$ as a validation set to aggregate them into a final classifier achieving the linear error rate.

For each subclass $\mathcal{H}_i$, if $S'$ is realizable by $\mathcal{H}$, let $h_i$ be the one-inclusion graph predictor with training sample $S'$, which satisfies $\mathbb{E}[\mathrm{er}_{P,f^*}(h_i)] \leq c\frac{d_i}{n}$ for some universal constant $c \in \mathbb{R}$; if $S'$ is not realizable by $\mathcal{H}_i$ we let $h_i$ be arbitrary. Furthermore, we only consider $h_i$ with $cd_i < n$ and let the remaining $h_i$ be arbitrary. This way we only have to compute finitely many one-inclusion graph predictors $h_i$, since $d_i \to \infty$.

Next we perform an online-to-batch conversion argument to aggregate these predictors. We fix an arbitrary order on $S''$ and run the weighted majority algorithm [Littlestone and Warmuth, 1994]. In particular, we let each $h_i$ be an expert with initial weight $w_i = \frac{1}{i(i+1)}$ and on mistake we multiply the weights by $e^{-\eta}$ for some $\eta \geq 2$. Denote by $M_i$ the number of mistakes of each expert $h_i$.[3]

The total number $M$ of mistakes of the weighted majority algorithm on $S''$ satisfies

$$M \leq \inf_i M_i + 1/\eta \ln(1/w_i) \leq \inf_i M_i + \ln(i+1) \leq M_{i^*} + \ln(i^*+1)$$

by a standard analysis (see, e.g., Corollary 3.1 by Cesa-Bianchi and Lugosi [2006]). Let $p = \mathrm{er}_{P,f^*}(h_{i^*})$, $\mu = pn/2$, and $\varepsilon \geq 2p$. Conditioned on $S'$, we have $M_{i^*} \leq \mathbb{E}[M_{i^*}] + \varepsilon n/2 = (p+\varepsilon)\frac{n}{2}$ with probability at least

$$1 - e^{-\frac{(\varepsilon/p)^2 \mu}{2+\varepsilon/p}} \geq 1 - e^{-\frac{(\varepsilon/p)^2 \mu}{2\varepsilon/p}} = 1 - e^{\varepsilon n/4}$$

by a (multiplicative) Chernoff bound. We thus have with the latter probability

$$M \leq \frac{n}{2}(\mathrm{er}_{P,f^*}(h_{i^*}) + \varepsilon) + \ln(i^*+1). \tag{1}$$

Let $h'_1, \ldots, h'_{n/2}$ be the weighted majority predictors in each step of the algorithm over $S''$.

**Claim.** *The expected error (conditioned on $S'$) of the (unweighted) majority vote $\hat{h} = \mathrm{Maj}(h'_1, \ldots, h'_{n/2})$ is $\mathcal{O}\left(\mathrm{er}_{P,f^*}(h_{i^*}) + \ln(i^*+1)/n\right)$.*

By Zhang [2005, Theorem 6 and Proposition 1], for $\varepsilon > 0$, with probability at least $1 - e^{-\varepsilon n/2}$, for some universal constant $c'$, we have

$$\frac{2}{n} \sum_{i=1}^{n/2} \mathrm{er}_{P,f^*}(h'_i) = \mathcal{O}\left(\frac{M}{n} + \sqrt{\frac{M}{n}\left(\frac{\log M}{n} + \varepsilon\right)} + \varepsilon\right) \leq c'(M/n + \varepsilon)$$

using the AM-GM inequality in the last step. For $\hat{h}$ this implies

$$\mathrm{er}_{P,f^*}(\hat{h}) \leq \mathop{\mathbb{E}}_{x \sim P}\left[\mathbb{1}\left[(2/n)\sum_{i=1}^{n/2}\mathbb{1}[h'_i(x) \neq f^*(x)] \geq \frac{1}{2}\right]\right]$$

$$\leq 2\mathop{\mathbb{E}}_{x \sim P}\left[(2/n)\sum_{i=1}^{n/2}\mathbb{1}[h'_i(x) \neq f^*(x)]\right] = 2\frac{2}{n}\sum_{i=1}^{n/2}\mathrm{er}_{P,f^*}(h'_i) \leq 2c'(M/n + \varepsilon).$$

Combining this with Equation (1), the expected error of $\hat{h}$ (conditioned on $S'$) is at most $3c'(\beta + \varepsilon)$ where $\beta = \mathrm{er}_{P,f^*}(h_{i^*}) + \ln(i^*+1)/n$ with probability at least $1 - 2e^{-\varepsilon n/4}$. The claim follows by

$$\mathop{\mathbb{E}}_{S''}[\mathrm{er}_{P,f^*}(\hat{h})|S'] = \int_{\alpha > 0} \Pr(\mathrm{er}_{P,f^*}(\hat{h}) > \alpha)d\alpha$$

$$= \int_{0 < \alpha \leq 3c'\beta} \Pr(\mathrm{er}_{P,f^*}(\hat{h}) > \alpha)d\alpha + \int_{\alpha > 3c'\beta} \Pr(\mathrm{er}_{P,f^*}(\hat{h}) > \alpha)d\alpha$$

$$\leq 3c'\beta + \int_{\alpha > 3c'\beta} 2e^{-\left(\frac{\alpha}{3c'} - \beta\right)n/4}d\alpha = \mathcal{O}\left(\beta + \frac{1}{n}\right).$$

Finally, as $h_{i^*}$ satisfies $\mathbb{E}[\mathrm{er}_{P,f^*}(h_{i^*})] = \mathcal{O}(d_{i^*}/n)$ ($S'$ is realizable by $\mathcal{H}_{i^*}$) and by the assumption $d_{i^*} \geq \ln(i^*+1)$, the expected error (conditioned on $S'$) of $\hat{h}$ is at most $\mathcal{O}(d_{i^*}/n)$. Taking the expectation over $S'$ and by the law of total expectation, the lemma follows.

$\square$

**Lemma 18.** *Let $\mathcal{H}$ be a class with $|\mathcal{H}| \geq 3$. The class $\mathcal{H}$ is not concept-nonuniformly learnable at rate faster than $1/n$.*

---

[3]To keep the set of experts finite, we can sum all experts with $cd_i \geq n$ and treat them as one.

# 5 Examples and broader perspective

In this section we discuss some examples and put the results into a broader perspective. For a rate $R$ and $T \in \{$uniform, universal, concept-nonuniform, marginal-nonuniform$\}$, we denote by $R$-$T$ the family of hypothesis classes that are $T$-learnable with rate $R$. By definition of learning with a fixed rate $R$ we have the following trivial inclusions:

$$^1/n\text{-uniform} \subseteq \begin{matrix} ^1/n\text{-marginal-nonuniform} \\ ^1/n\text{-concept-nonuniform} \end{matrix} \subseteq {}^1/n\text{-universal}.$$

Our results yield that the inclusion "$^1/n$-uniform $\subseteq {}^1/n$-marginal-nonuniform" is actually an equality. This is the case as the first family is characterized by the finiteness of the VC dimension, the latter family is characterized by the finiteness of the VC-eluder dimension (Theorem 3), and both parameters are finite together [Hanneke and Xu, 2024, Remark 19]. Then, as each class that is concept-nonuniformly learnable with linear rate is a countable union of VC classes (Theorem 16) we also have the inclusion "$^1/n$-marginal-nonuniform $\subseteq {}^1/n$-concept-nonuniform". The next example shows that this is indeed a strict inclusion.

**Example 1** ($^1/n$-concept-nonuniform but not $^1/n$-marginal-nonuniform). *Let $\mathcal{H} = \{A \subseteq \mathbb{N} \mid |A| < \infty\}$. Note that $\mathrm{VC}(\mathcal{H}) = \mathrm{VCE}(\mathcal{H}) = \infty$, which means that $\mathcal{H}$ is not marginal-nonuniformly learnable with linear rate and not uniformly learnable. However, it can be easily seen that $\mathcal{H} = \bigcup_{i=1}^{\infty} \mathcal{H}_i$ with $\mathcal{H}_i = \{A \subseteq \mathbb{N} \mid |A| \leq i\}$ and $\mathrm{VC}(\mathcal{H}_i) = i$. Thus $\mathcal{H}$ is concept-nonuniformly learnable with rate $^1/n$. Furthermore, this class is countable and hence marginal-nonuniformly learnable with arbitrarily slow rates, see Benedek and Itai [1991] and Section 6.*

Additionally, by definition we have (with m.nu. = marginal-nonuniform):

$$e^{-n}\text{-m.nu.} \subseteq {}^1/n\text{-m.nu.} \subseteq \text{arbitrarily slow-m.nu.} \subseteq \text{arbitrarily slow-universal}.$$

The first inclusion is strict, as the first family contains exactly all finite hypothesis classes, while the second family contains all VC classes (Theorem 3). The other two inclusions are also strict as shown in the next two examples.

**Example 2** (Marginal-nonuniform with arbitrarily slow rate but not $^1/n$-universal learnable). *Let $X = \mathbb{N}$ and $\mathcal{H} = 2^{\mathbb{N}}$. Ben-David et al. [1995] showed that this class is marginal-nonuniformly learnable (independently of rates) but not concept-nonuniformly learnable. Moreover, as this class shatters a countable set, it is not universally learnable with linear rate, see Bousquet et al. [2021],*

**Example 3** (Only arbitrarily slow universal). *Let $X = [0,1]$ and $\mathcal{H}$ the set of all open sets over $X$. Ben-David et al. [1995] showed that this class is universally learnable with arbitrarily slow rate, but neither marginal nor concept-nonuniformly learnable. It is easy to see that this class also shatters a countable, and is, thus, not universally learnable with linear rate (see Bousquet et al. [2021]).*

There are also classes that are concept-nonuniformly but not marginal-nonuniformly learnable.

**Example 4** (Concept-nonuniform but not marginal-nonuniform). *Let $X = (0,1)$, $S$ be the set of all sub-intervals with rational endpoints in $X$, and $\mathcal{H}$ be all finite unions of $S$. Ben-David et al. [1995] showed that this class is concept-nonuniformly learnable (and thus with linear rate) but not marginal-nonuniformly learnable (independently of rates).*

**Example 5** (Linear universal but neither concept-nonuniform nor marginal-nonuniform). *Let $X_1 = \{A \subseteq \mathbb{R} \mid |A| < \infty\}$ and $\mathcal{H}_1 = \{h_y \mid y \in \mathbb{R}\}$ with $h_y(S) = \mathbb{1}[y \in S]$. Bousquet et al. [2021] (Example 2.7) argued that $\mathcal{H}_1$ is not representable as a countable union of VC classes, thus not concept-nonuniformly learnable. It is universally learnable with exponential rate however (and thus also linear rate). Let $\mathcal{H}_2$ be from Example 4, which is not marginal-nonuniformly learnable. As $\mathcal{H}_2$ is concept-nonuniformly learnable at linear rate it is also universally learnable at linear rate. Take the disjoint union $\mathcal{H} = \mathcal{H}_1 \cup \mathcal{H}_2$. It is easy to see that $\mathcal{H}$ is still universally learnable with linear rate but neither concept nor marginal-nonuniformly learnable (independently of rates).*

The only remaining open case is whether there exists a class that is not concept-nonuniformly learnable, is universally learnable with linear rate, and marginal-nonuniformly learnable at arbitrarily slow rate. Except this case, this shows all possible inclusions of the rates in marginal and concept-nonuniform learning, and their relationship to uniform and universal learning. Figure 1 gives an overview with the missing case marked as "?".

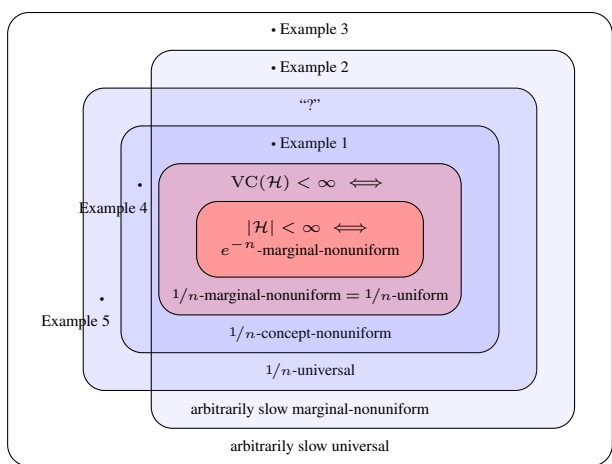

Figure 1: Overview on the learning rates related to marginal and concept-nonuniform learning.

## 6 Discussion and future work

We end with some discussion on related aspects and some potential future work.

**Weaker notions of realizability.** Bousquet et al. [2021] considered a weaker notion of realizability: a distribution $P_{XY}$ over $X \times Y$ is realizable if $\inf_{h \in \mathcal{H}} \mathrm{er}_{P_{XY}}(h) = 0$ where $\mathrm{er}_{P_{XY}}(h) = \Pr_{(x,y) \sim P_{XY}}(h(x) \neq y)$. Any such distribution $P$ has $P(y = 1|x) \in \{0, 1\}$ for almost all $x$. Hence we can more generally say that a marginal distribution $P$ over $X$ and a target $f^* \in \{0, 1\}^X$ is realizable if $\inf_{h \in \mathcal{H}} \mathrm{er}_{P, f^*}(h) = 0$. This allows slightly more general definitions of marginal and concept-nonuniform learning, with again constants depending on either $P$ or $f^*$. Our proofs for the marginal-nonuniform case can be adapted to the weakly realizable case. Thus, the landscape of potential learning rates stays exactly the same, independently of weak or strict realizability. This is also the case in uniform learning. However, for universal and concept-nonuniform learning the two variants of realizability result in different learnable classes. Characterizing the differences is open.

**Universal Glivenko-Cantelli does not determine learnability.** The problem of distinguishing a marginal-nonuniformly learnable class with arbitrarily slow rates from a not learnable class is still open. A natural candidate is the universal Glivenko-Cantelli property, which is the analogue of uniform convergence in the context of universal learning [Bousquet et al., 2021, van Handel, 2013]. However, the class $\{A \subseteq \mathbb{R} \mid |A| < \infty\}$ does not have the Glivenko-Cantelli property [van Handel, 2013], but is marginal and concept-nonuniformly learnable [Ben-David et al., 1995].

**Marginal-nonuniform learning on countable domains.** Benedek and Itai [1991] show that discrete distributions are fixed-distribution learnable and thus all classes on countable domains are marginal-nonuniformly learnable. However, our results show that the family of all such classes require at least arbitrarily slow rates. Thus, while all classes on countable domains are marginal-nonuniformly learnable (independently of rate), there is no single rate capturing the learnability.

**Marginal-nonuniform $\cap$ concept-nonuniform vs. uniform learning.** We showed that in the case of linear rates, marginal-nonuniform learning collapses to uniform learning. Interestingly, if we consider learning independently of rates a different situation arises. The intersection of marginal-nonuniform and concept-nonuniform is then a strict superset of uniformly learnable classes, see Ben-David et al. [1995]. In fact Example 1, the concept class of all finite subsets of $\mathbb{N}$, is in the intersection of the two nonuniform settings but not uniformly learnable.

**Other learning settings.** Universal rates were also considered for active [Hanneke et al., 2024] and online learning [Blanchard, 2022]. Marginal-nonuniform rates could be interesting here, as well. For example, typical worst-case lower bounds in active learning for linear classifiers, are achieved by fixed marginal distributions [Balcan, Hanneke, and Vaughan, 2010, Hanneke et al., 2024].

## Acknowledgments and Disclosure of Funding

SM is supported by a Robert J. Shillman Fellowship, by ISF grant 1225/20, by BSF grant 2018385, by an Azrieli Faculty Fellowship, by Israel PBC-VATAT, and by the Technion Center for Machine Learning and Intelligent Systems (MLIS), and by the European Union (ERC, GENERALIZATION, 101039692). Views and opinions expressed are however those of the author(s) only and do not necessarily reflect those of the European Union or the European Research Council Executive Agency. Neither the European Union nor the granting authority can be held responsible for them.

MT acknowledges support from a DOC fellowship of the Austrian Academy of Sciences (ÖAW).

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

## A Marginal nonuniform

**Lemma 4.** *Any class $\mathcal{H}$ with $3 \leq |\mathcal{H}| < \infty$ is marginal-nonuniformly learnable with exact rate $e^{-n}$.*

*Proof.* We start with the upper bound for all finite classes. The standard analysis of PAC learning with finite $\mathcal{H}$ (see e.g., Example 8 of Hanneke and Xu [2024]) yields for any fixed $P \in \Delta(X)$, $f^* \in \mathcal{H}$:

$$\mathbb{E}[\mathrm{er}_{P,f^*}(\hat{h}_n)] \leq |\mathcal{H}| \exp\left(-\left(\min_{h \in \mathcal{H}, \mathrm{er}_{P,f^*}(h)>0} \mathrm{er}_{P,f^*}(h)\right) n\right).$$

Now as $\mathcal{H}$ is finite and by taking the maximum over $f^* \in \mathcal{H}$ we get for some constant $c$:

$$\mathbb{E}[\mathrm{er}_{P,f^*}(\hat{h}_n)] \leq |\mathcal{H}| \exp\left(-\left(\min_{h \in \mathcal{H}, \mathrm{er}_{P,f^*}(h)>0} \mathrm{er}_{P,f^*}(h)\right) n\right)$$

$$\leq |\mathcal{H}| \max_{h' \in \mathcal{H}} \exp\left(-\left(\min_{h \in \mathcal{H}, \mathrm{er}_{P,h'}(h)>0} \mathrm{er}_{P,h'}(h)\right) n\right)$$

$$\leq |\mathcal{H}| \exp(-cn).$$

Note that the constant $c$ is independent of any concept $h' \in \mathcal{H}$ (or rather it is the worst case over the finitely many $h'$) and thus only depends on the marginal $P$.

We continue with the lower bound. A finite class $\mathcal{H}$ with $|\mathcal{H}| \geq 3$ is not universally learnable at rate faster than $e^{-n}$, see Bousquet et al. [2021], Schuurmans [1997]. Thus, this is also not possible in the stricter marginal-nonuniform case. $\square$

**Lemma 5.** *Every class $\mathcal{H}$ with finite VC-dimension is marginal-nonuniformly learnable at rate $1/n$.*

*Proof.* The class $\mathcal{H}$ is uniformly learnable at rate $1/n$ (using the one-inclusion graph algorithm [Haussler et al., 1994]). Thus, it is also learnable at this rate in the less strict marginal-nonuniform setting. $\square$

**Lemma 6.** *Every infinite class $\mathcal{H}$ is not marginal-nonuniformly learnable at rate faster than $1/n$.*

*Proof.* We adapt the proof of Lemma 14 by Hanneke and Xu [2024]. As $\mathcal{H}$ has infinite size it has an infinite eluder sequence [Hanneke and Xu, 2024, Lemma 8]. Let $E = \{(x_1, y_1), (x_2, y_2), \dots\}$ be such a sequence. Define the following marginal distribution $P$ on $X$ with $P(x_i) = 2^{-i}$ for all $i \in \mathbb{N}$.

Let $\overline{S}_n = \{s_1, \dots, s_n\} \sim P^n$ be an unlabeled iid sample of size $n \geq 2$. For $t = \lceil \log n \rceil$, the probability that $\overline{S}_n$ does not contain any point in $\{x_i : i > t\}$ is

$$\Pr(\overline{S}_n \cap \{x_i, i > t\} = \emptyset) = \prod_{i=1}^{n} \Pr(s_i \in \{x_1, \dots, x_t\}) = (1 - 2^{-t})^n \geq (1 - 1/n)^n \geq 1/4. \quad (2)$$

Let $S_n = \{(s_1, y_1), \dots, (s_n, y_n)\}$ be a labeled (realizable) sample and $\hat{h}_n = A(S_n)$. As $E$ is an eluder sequence, there are concepts $h, h' \in \mathcal{H}$ with $h(x_i) = h'(x_i) = y_i$ for all $i \in [t]$ and $h(x_{t+1}) \neq h'(x_{t+1})$. Let the target concept $f^*$ be chosen uniformly at random from $\{h, h'\}$. For any deterministic learner $A$ under the event that $\overline{S}_n \cap \{x_i : i > t\} = \emptyset$ we have

$$\mathrm{er}_{P,f^*}(A(S_n)) \geq \frac{1}{2} 2^{-(t+1)} \geq \frac{1}{4n}. \quad (3)$$

This holds as $x_{t+1}$ is drawn with probability $2^{-(t+1)}$ and as $x_{t+1} \notin \overline{S}_n$ the learner $A$ will err with probability $1/2$ as the true label of $x_{t+1}$ is drawn uniformly at random from $\{0, 1\}$. By combining Equation (2) and Equation (3) we get

$$\mathbb{E}_{S_n, f^*}[\mathrm{er}_{P,f^*}(A(S_n))] \geq \frac{1}{4} \frac{1}{4n} = \frac{1}{16n}.$$

Note that the uniform distribution over $h, h'$ yields a lower bound of $1/4n$ simultaneously for all deterministic learners. Thus, by e.g., Yao's principle, we know that for any randomized learner, either $h$ or $h'$ gives deterministically the same lower bound, proving the claim. $\square$

### A.1 Arbitrarily slow rates

We use here the notion of an infinite VC-eluder sequence, see, e.g., Hanneke and Xu [2024]. Let $\mathcal{H}$ be a class and $n_k = \binom{k}{2}$ for all $k \in \mathbb{N}$. A sequence $S = \{(x_1, h(x_1)), (x_2, h(x_2)), \dots\}$ is an *infinite VC-eluder sequence* (centered at $h$) if for all $k \in \mathbb{N}$, the set $\{x_{n_k+1}, \dots, x_{n_k+k}\}$ is shattered by the version space $\mathrm{VS}(S_{\leq n_k}, \mathcal{H})$. The class $\mathcal{H}$ has an infinite VC-eluder sequence if and only if $\mathrm{VCE}(\mathcal{H}) = \infty$ [Hanneke and Xu, 2024].

Moreover, we rely on the following technical lemma by Bousquet et al. [2021].

**Lemma 19** (Lemma 5.12, Bousquet et al. 2021)**.** *For every rate function $R$, there exists probabilities $\{p_t\}_{t \in \mathbb{N}}$ with $\sum_{t \in \mathbb{N}} p_t = 1$, two increasing sequences of positive integers $\{n_t\}_{t \in \mathbb{N}}$, $\{k_t\}_{t \in \mathbb{N}}$, and a constant $1/2 \leq C \leq 1$ such that all the following hold:*

1. $\sum_{k \; k_t} p_k \leq \frac{1}{n_t}$,

2. $n_t p_{k_t} \leq k_t$, *and*

3. $p_{k_t} = CR(n_t)$.

**Lemma 13.** *Let $\mathcal{H}$ be a class with infinite VC-eluder dimension. Marginal-nonuniformly learning $\mathcal{H}$ requires at least arbitrarily slow rates.*

*Proof.* We adapt the proof of Lemma 7 by Hanneke and Xu [2024]. As $\mathrm{VCE}(\mathcal{H}) = \infty$, the class $\mathcal{H}$ has an infinite VC-eluder sequence $E = \{(x_1, f'(x_1)), (x_2, f'(x_2)), \dots\}$ for some $f' \in \mathcal{H}$. Let $X_k = \{x_{n_k+1}, \dots, x_{n_k+k}\}$ be the shattered sets of increasing size in $E$ for all $k \in \mathbb{N}$. We define the following marginal distribution $P \in \Delta(X)$. For all all $k \in \mathbb{N}$, let $\Pr(x \in X_k) = p_k$ and $\Pr(x) = p_k/k$ for all $x \in X_k$, where the sequence $\{p_k\}_{k \in \mathbb{N}}$ will be specified later. Let $R$ be an arbitrary rate function. We will show that no learner can marginal-nonuniformly learn under $P$ faster than rate $R$.

Let $S_n$ be an unlabeled iid sample from $P$ of size $n$. Let $\{k_t\}_{t \in \mathbb{N}}$ be an increasing sequence of positive integers to be specified later. Denote $X_{k>t} = \bigcup_{t'>t} X_{k_{t'}}$ for all $t \in \mathbb{N}$. For all $t \in \mathbb{N}$, $j \in [k_t]$, and $k$ we define the event

$$A_{n,k,t,j} = \left\{ S_n \cap (X_{k>t} \cup \{x_{n_{k_t}+j}\}) = \emptyset \right\}.$$

Note that, $A_{n,k,t,j}$ indicates whether the sample $S_n$ contains no points from any $X_{k_{t'}}$ (for $t' > t$) nor the point $x_{n_{k_t}+j} \in X_{k_t}$. That is, by definition of an infinite VC-eluder sequences, if $A_{n,k,t,j}$ holds, the version space under the sample $S_n$ labeled by $f'$ contains a hypothesis $f'' \in \mathcal{H}$ that classifies $x_{n_{k_t}+j}$ differently, i.e., $f'(x_{n_{k_t}+j}) \neq f''(x_{n_{k_t}+j})$. This holds because $X_{k_t}$ is shatterable in this case and $x_{n_{k_t}+j} \in X_{k_t}$ is not in the sample. Note that $f''$ can depend on $n$. Consider an algorithm $\hat{h}_n$ and its prediction on $x_{n_{k_t}+j}$ given by sample $S_n$ (labeled by $f'$ (or equally $f''$) restricted to $S_n$). Select the ground truth $f^* \in \{f', f''\}$ as the one hypothesis whose label $f^*(x_{n_{k_t}+j})$ is the more unlikely one to be predicted by the classifier returned by the learner. Thus, if $A_{n,k,t,j}$ holds we have that $\mathrm{er}_{P,f^*}(\hat{h}_n) \geq \frac{1}{2}\frac{p_{k_t}}{k_t}$, as $x_{n_{k_t}+j}$ will be drawn with the probability $\frac{p_{k_t}}{k_t}$ and then misclassified with probability at least $1/2$. Also note that $\Pr(A_{n,k,t,j}) = (1 - \sum_{k>k_t} p_k - p_{k_t}/k_t)^n$.

We now apply Lemma 19 to get the $\{p_t\}_{t \in \mathbb{N}}$, $\{k_t\}_{t \in \mathbb{N}}$, and $\{n_t\}_{t \in \mathbb{N}}$ sequences, and $C$ with the properties as stated. Then we get for all $t \in \mathbb{N}$ with $n_t \geq 3$ (and thus for infinitely many $n \in \mathbb{N}$):

$$\mathbb{E}[\mathrm{er}_{p,f^*}(\hat{h}_{n_t})] \geq \sum_{j \in [k_t]} \frac{p_{k_t}}{2k_t} \Pr(A_{n_t,k,t,j}) \geq \frac{p_{k_t}}{2}(1 - \sum_{k>k_t} p_k - p_{k_t}/k_t)^{n_t}$$

$$\geq \frac{p_{k_t}}{2}\left(1 - \frac{2}{n_t}\right)^{n_t}$$

$$\geq \frac{p_{k_t}}{54} \geq \frac{C}{54} R(n_t).$$

$\square$

## A.2 Fine-grained linear rates through finite VC-eluder dimension

**Lemma 12.** *There exists an absolute constant $\beta$ such that for each class $\mathcal{H}$ with $\mathrm{VCE}(\mathcal{H}) < \infty$, for all learners $\hat{h}_n$ there is a marginal distribution $P \in \Delta(X)$, such that for infinitely many $n \in \mathbb{N}$ there exists a target concept $f_n^*$ with*

$$\mathbb{E}[\mathrm{er}_{P,f_n^*}(\hat{h}_n)] \geq \beta \frac{\mathrm{VCE}(\mathcal{H})}{n} .$$

*Proof.* The proof is similar to Lemma 13 but uses rate $d/n$ instead of arbitrarily slow ones. Let $\mathrm{VCE}(\mathcal{H}) = d$. The class $\mathcal{H}$ has an infinite $d$-VC-eluder sequence $S = \{(x_1, f^*(x_1)), (x_2, f^*(x_2)), \dots\}$ for some $f^* \in \mathcal{H}$. Let for all $k \in \mathbb{N}$ the $X_k = \{x_{kd-d+1}, \dots, x_{kd}\}$ be the shattered sets of size $d$ in $S$. We define the following marginal distribution $P \in \Delta(X)$. For all $k \in \mathbb{N}$, let $\Pr(x \in X_k) = p_k$ and $\Pr(x) = p_k/d$ for all $x \in X_k$, where the sequence $\{p_k\}_{k \in \mathbb{N}}$ will be specified later. Let $R(n) = d/n$. We will show that no learner can marginal-nonuniformly learn under $P$ faster than rate $R$.

Let $S_n$ be an unlabeled iid sample from $P$ of size $n$. Let $\{k_t\}_{t \in \mathbb{N}}$ be an increasing sequence of positive integers to be specified later. Denote $X_{>k} = \bigcup_{j>k} X_j$ for all $k \in \mathbb{N}$. For all $t \in \mathbb{N}$, $j \in [d]$, and $k$ we define the event

$$A_{n,k,j} = \left\{ S_n \cap (X_{>k} \cup \{x_{kd-d+j}\}) = \emptyset \right\} .$$

Note that, $A_{n,k,j}$ indicates whether the sample $S_n$ contains no points from any $X_j$ (for $j > k$) nor the point $x_{kd-d+j} \in X_k$. That is, by definition of an infinite $d$-VC-eluder sequences, if $A_{n,k,j}$ holds, the version space under the sample $S_n$ labeled by $f'$ contains a hypothesis $f'' \in \mathcal{H}$ that classifies $x_{kd-d+j}$ differently, i.e., $f'(x_{kd-d+j}) \neq f''(x_{kd-d+j})$. This holds because $X_k$ is shatterable in this case and $x_{kd-d+j} \in X_k$ is not in the sample. Note that $f''$ can depend on $n$. Consider an algorithm $\hat{h}_n$ and its prediction on $x_{kd-d+j}$ given by sample $S_n$ (labeled by $f'$ (or equally $f''$) restricted to $S_n$). Select the ground truth $f^* \in \{f', f''\}$ as the one hypothesis whose label $f^*(x_{kd-d+j})$ is the more unlikely one to be predicted by the classifier returned by the learner. Thus, if $A_{n,k,j}$ holds we have that $\mathrm{er}_{P,f^*}(\hat{h}_n) \geq \frac{1}{2}\frac{p_k}{d}$, as $x_{kd-d+j}$ will be drawn with the probability $\frac{p_k}{d}$ and then misclassified with probability at least $1/2$. Also note that $\Pr(A_{n,k,j}) = (1 - \sum_{j>k} p_j - p_j/d)^n$.

We now apply Lemma 19 to get the $p_t$, $k_t$, and $n_t$ sequences, and $C$ with the properties as stated (we do not use the $k_t$ sequence here). Then we get for all $t \in \mathbb{N}$ with $n_t \geq 3$ (and thus for infinitely many $n \in \mathbb{N}$):

$$\mathbb{E}[\mathrm{er}_{p,f^*}(\hat{h}_{n_t})] \geq \sum_{j \in [d]} \frac{p_k}{2d} \Pr(A_{n_t,k,j}) \geq \frac{p_{k_t}}{2}(1 - \sum_{k>k_t} p_k - p_k/d)^{n_t}$$

$$\geq \frac{p_{k_t}}{2}\left(1 - \frac{2}{n_t}\right)^{n_t}$$

$$\geq \frac{p_{k_t}}{54} \geq \frac{C}{54}\frac{d}{n_t} .$$

$\square$

# B  Concept nonuniform

**Lemma 18.** *Let $\mathcal{H}$ be a class with $|\mathcal{H}| \geq 3$. The class $\mathcal{H}$ is not concept-nonuniformly learnable at rate faster than $1/n$.*

*Proof.* This is an adaptation of standard lower bounds for PAC learning, see, e.g., Blumer, Ehrenfeucht, Haussler, and Warmuth [1989], Anthony and Bartlett [1999]. As $|\mathcal{H}| \geq 3$ there exist two hypotheses $h, h'$ and two points $x, x' \in X$ such that $h(x) = h'(x)$ and $h(x') \neq h'(x')$. We will show that at least one of the two hypotheses requires a rate of $1/n$ to be concept-nonuniformly learnable. Let $n \in \mathbb{N}$ be a sample size with $n \geq 2$ and let the marginal distribution $P \in \Delta(X)$ be given as $P(x) = 1 - \frac{1}{n}$ and $P(x') = \frac{1}{n}$. Let $\bar{S}_n$ be an unlabelled iid sample from $P^n$ and denote the event

that $\bar{S}_n$ does not contain $x'$ as $A$. Note that

$$\Pr_{\bar{S}_n \sim P^n}(A) = \left(1 - \frac{1}{n}\right)^n \geq \left(1 - \frac{1}{2}\right)^2 = 1/4.$$

Now, for each $n \in \mathbb{N}$, let $\hat{h}_n$ be a classifier returned by a learning algorithm given the sample $\bar{S}_n$ labelled by $f^* \in \{h, h'\}$, where the latter is to be determined shortly. For each $n \in \mathbb{N}$, let $y_n \in \{0, 1\}$ be the label that is less likely to be predicted by (the potentially randomized) $\hat{h}_n$ for $x'$. At least one of $y_n = 0$ or $y_n = 1$ appears infinitely often in the sequence $(y_n)_n$. Call this label $y^*$. We select $f^*$ as the hypothesis with $f^*(x') = y^*$. Consider the sample under the event $A$, i.e., the sample contains $n$ times the point $x$. For infinitely many $n$, under the event $A$, the classifier $\hat{h}_n$ will have an expected error of at least $\frac{1}{n}\frac{1}{2}$. Indeed, the point $x'$ will be drawn with probability $1/n$ and for infinitely many $n$ the classifier makes an error with probability at least $1/2$ (by the choice of $f^*$). Thus, overall the error probability of $\hat{h}_n$ for the target concept $f^*$ satisfies

$$\mathrm{er}_{P,f^*}(\hat{h}_n) \geq \frac{1}{2n}P(A) \geq \frac{1}{8n}$$

for infinitely many $n$. The claim follows. $\qquad\square$

