# OpenReview forum: "Marginal-Nonuniform PAC Learnability"
_NeurIPS.cc/2025/Conference — NeurIPS 2025 poster_

### Official Review · Reviewer_NqaX · 2025-06-11

**Clarity:** 3
**Significance:** 2
**Originality:** 3
**Rating:** 5
**Confidence:** 2

**Summary:**

This paper considers the problem of marginal-nonuniform PAC learning. Here there is a known class C of functions on a space X. One wants to (realizably) PAC learn an unknown function f in C on an unknown distribution D on X, but in a way where the rate at which the error converges to 0 in terms of samples is allowed to depend on the marginal distribution D.

In particular, they say that you can learn with rate R (for some function R going monotonly to 0) if there is an algorithm A so that after receiving n samples (x,y) where x ~ D and y = f(x) for some f in C returns a hypothesis h with error over C at most O_D( R( Omega_D(n) ) ). In other words, when run with any fixed marginal D, the error goes to 0 at a rate with the same functional form as R.

The main result of this paper is a classification of achievable rates. In particular:
1) In some trivial cases, one can learn in <= 1 sample and take R = 0.
2) Otherwise, if C is finite, one learns with exponential rate R(n) = e^{-n}.
3) Otherwise, if C has finite VC dimension, one learns with rate R(n) = 1/n.
4) Finally, if C has infinite VC dimension, then there is no rate R(n) that goes to 0 so that we can learn with rate R.

They also show that in case (3) the learning rate can be taken more precisely to be bounded by O(d/n) where d is the VC-eluder-dimension.

**Questions:**

* Is the proof of Theorem 16 essentially just contained in the citations given? If not, what is the proof?
* What *was* previously known about this problem?
* What in general do you mean by considering fine-grained rates?

**Ethical Concerns:**

["NO or VERY MINOR ethics concerns only"]

**Final Justification:**

It's a solid paper. I am not quite sure that it meets the threshold of "Technically solid paper, with high impact on at least one sub-area of AI or moderate-to-high impact on more than one area of AI" because the result seems a little specialized.

**Limitations:**

yes

**Quality:**

3

**Strengths And Weaknesses:**

Strengths:
* The basic idea of marginal-nonuniform learning is a nice one and this paper essentially solves some version of it.
* The final classification is clean and not entirely expected.
* The proofs are easy to follow.
* Generally the background is well explained.

Weaknesses:
* The technical definition of this kind of marginal-nonuniform learning in terms of rate functions R is a bit complicated and not especially natural.
* The results here are all relatively easy and do not seem to introduce technical tools that are likely to have broader applicability. (Though the VCE stuff has some more complicated ideas)

Comments:
* Line 91: *What* is a strict version of the realizability assumption? My initial read was that you were claiming that the function er was.
* Definition 2:
 - Isn't h_n^ more naturally the (randomized) output of an algorithm rather than the algorithm itself?
 - What do you mean by "a faster rule than rate R"?

---

> ### Author Rebuttal · Authors · 2025-07-30
>
> Thanks for the feedback.
>
> > The technical definition of this kind of marginal-nonuniform learning in terms of rate functions R is a bit complicated and not especially natural.
>
> We will provide more exposition in the revised version. Intuitively our error rates allow a dependence on the marginal distribution generating the unlabeled data points. In the context of universal PAC learning (Bousquet et al., 2021) our definition (Definition 2) is the natural variant of an error rate with constants depending on just the marginal distribution instead of the full distribution over $X\times Y$ (as in universal learning).
>
> > The results here are all relatively easy and do not seem to introduce technical tools that are likely to have broader applicability.
>
> We politely disagree. We believe that identifying the right structure (VCE-dimension) for marginal nonuniform learning is non-trivial and valuable. After considerable technical difficulties this then lead to our rate trichotomy. In particular, our result on fine-grained learnability with rate $\operatorname{VCE}(H)/n$ requires novel ideas to show that any version space will have eventually finite VC dimension (with probability $1$) relying on König’s Lemma. This might be of interest for other learning settings, as well. We will clarify our exposition in the revised version and add a proof sketch of this main result, which is currently only in the appendix.
>
> > Line 91: What is a strict version of the realizability assumption? My initial read was that you were claiming that the function er was.
>
> We call it "strict", since we assume that $f^*\in\mathcal H$. That is in contrast to the (weaker) realizability notion of Bousquet et al. (2021) in unviersal learning. Basically we take the minimum instead of an infimum, see Section 6 for more details.
>
> > Isn't h_n^ more naturally the (randomized) output of an algorithm rather than the algorithm itself?
>
> We choose to follow the notation in universal PAC learning (Bousquet et al., 2021 and follow up works), which slightly conflates the algorithm and the output hypothesis. The reason is that writing $A(S)$ for the output hypothesis requires explicitly stating/defining the learning algorithm $A$, which can be a bit convoluted sometimes.
>
> > What do you mean by "a faster rule than rate R"?
>
> If you mean “rate faster than rate R” then please see Definition 2.2. This follows similar definitions in universal learning (Bousquet et al. 2021).
>
> > Is the proof of Theorem 16 essentially just contained in the citations given? If not, what is the proof?
>
> No. Previous results show that a class is concept-nonuniformly learnable (ignoring rates) if and only if it is a countable union of VC classes (Lemma 15, which is proven by Benedek and Itai (1994)). The best known rate was $\log(n)/n$. We show that the optimal rate is $1/n$, which is achievable whenever the class is concept-nonuniformly learnable at all. The proof is in the appendix.
>
> > What was previously known about this problem?
>
> If you mean concept-nonuniform learnability see the above comment. If you instead mean marginal-nonuniform learnability then not much was known besides the definition itself and certain relationships with related learning settings (see Ben-David et al., 1995). We are the first to obtain the three possible cases and the exact rates.
>
> > What in general do you mean by considering fine-grained rates?
>
> By fine-grained rates, we refer to tighter bounds that offer sharper control over the tails of the learning curves—that is, in the regime of large sample sizes. In both this paper and earlier work by Bousquet et al (2023), the bounds are distribution-dependent: for each distribution $D$, the bound may begin to apply only after a certain threshold $N(D)$, but once it does, it is significantly sharper than the counterpart coarse-grained counterparts. For example, in this work, the VC-eluder dimension (VCE) replaces the VC dimension in the linear-rate regime, yielding potentially much tighter rates when VCE is substantially smaller than VC.

---

> > ### Comment · Reviewer_NqaX · 2025-08-01
> > **Rebuttal Reply**
> >
> > (1) The VCE dimension for the fine grained result seems a bit more substantial (though the introduction should have a techniques section that highlights your main ideas and any that you consider innovative). Are there any substantial ideas in the proof of the main classification result though?
> >
> > (2) In line 90-91 you say "this is a strict version of the realizability assumption", but it is not at all clear from context what the "this" in the sentence refers to. You should clarify this.
> >
> > (3) If you are referring to h_n^ as the algorithm even if it is not, you need to at least clearly state in the paper that this is what you are doing so as to not lead to confusion.
> >
> > (4) Where exactly is the proof of Theorem 16? The paper does not seem to have an Appendix. Furthermore, if the proof is to be deferred, you need to include a forward reference to where the proof is to be found.

---

> > > ### Author Response · Authors · 2025-08-04
> > >
> > > Thanks for the follow-up comments.
> > >
> > > > Comment: (1) The VCE dimension for the fine grained result seems a bit more substantial (though the introduction should have a techniques section that highlights your main ideas and any that you consider innovative). Are there any substantial ideas in the proof of the main classification result though?
> > >
> > > Thanks for the suggestion. We will expand our main contribution paragraph in the introduction with an overview on techniques.
> > >
> > > Regarding the classification results: We think that the most meaningful part of the proof lies in the lower bounds—namely, showing that every infinite class has at least linear convergence rate, and that for classes with infinite VC dimension, the rate can be made arbitrarily slow. This is somewhat surprising as one might expect that faster than linear rates from uniform PAC learning are possible by allowing marginal-dependent constants. The upper bounds in the classification result are more direct: the linear rate for finite VC dimension follows from the uniform setting (though our fine-grained version using the VC-eluder dimension gives a significant refinement), and the exponential rate for finite classes is relatively simple to establish.
> > >
> > > > (2) In line 90-91 you say "this is a strict version of the realizability assumption", but it is not at all clear from context what the "this" in the sentence refers to. You should clarify this.
> > >
> > > We agree that "this" is not ideal here. We will improve this part. With "this" we refer here to the fact from the sentence before that the (deterministic) ground truth is part of the class $f^*\in\mathcal{H}$. We call this strict realizability in contrast to the (weaker) realizability notion of Bousquet et al. (2021), which only assumes that $\inf_{h\in\mathcal{H}} \operatorname{er}_{P}(h)=0$ for the underlying distribution $P$ over $X\times Y$.
> > >
> > > > (3) If you are referring to h_n^ as the algorithm even if it is not, you need to at least clearly state in the paper that this is what you are doing so as to not lead to confusion.
> > >
> > > We will clarify this notation in the revised version. The two different usages come from writing $\exists \hat h_n$, by which we mean there exists an algorithm that outputs $\hat h_n$, and writing $\operatorname{er}(\hat h_n)$, by which we mean the true error of the output hypothesis.
> > >
> > > > (4) Where exactly is the proof of Theorem 16? The paper does not seem to have an Appendix. Furthermore, if the proof is to be deferred, you need to include a forward reference to where the proof is to be found.
> > >
> > > You can find the appendix in the supplementary material. In the revised version we will add proof sketches and point more explicitly to missing proofs.

---

### Official Review · Reviewer_ycGG · 2025-06-19

**Clarity:** 4
**Significance:** 3
**Originality:** 3
**Rating:** 4
**Confidence:** 3

**Summary:**

This paper revisits the classical model of non-uniform PAC learning, where the error convergence rate includes constants depend on the true concept or the data distribution. The main conclusion of this paper is that when considering marginal-nonuniform learning (case that the constant allows to be dependent on feature distribution but is required to be uniform across concepts), the optimal rate is a trichonomy, characterized by the VC-eluder dimension. This reveals some inclusions regarding uniform learning at rate $1/n$ and some non-uniform rates.

**Questions:**

- I am concerning the significance and applicability of theoretically studying the non-uniform settings of learning \emph{in the present days}. Specifically, casting at its greatest generality, what conclusions can be drawn from research in this direction, and what problems can be solved? How does resolving these issues promote the study of machine learning theory or assist in solving real-world machine learning problems? I appreciate that this is a fundamental research area that worth to investigate, but to be honest, such concern affect my score to some extent (and sadly, I can hardly tell a clear criteria of how this affects). I would appreciate it if the authors could share some opinions at this point.
 - In line 106-107 you've mentioned that "there are families of classes such that $VCE(\mathcal{H})=1$", while $VC(\mathcal{H})$ is unbounded", while in line 135-136 you said that $1\leq VCE(\mathcal{H}) <\infty$ is equivalent to $|\mathcal{H}|=\infty$ and $VC(\mathcal{H})<\infty$. Do these two statements mean that while there exists classes $\mathcal{H}$ with VCE-dimension $1$ and *arbitrarily large* VC-dimension, there does not exist a class $\mathcal{H}$ with VCE-dimension $1$ and *infinite* VC-dimension?

**Ethical Concerns:**

["NO or VERY MINOR ethics concerns only"]

**Final Justification:**

Thanks the authors for the detailed response. My concern has been properly resolved, and specifically, it convinced me that the paper fits within the borader theme of beyond-the-worst-case generalization analysis, which definitely serves as one of the necessary directions that push forward the study of learning theory. Therefore, I have updated the review and kept my score.

**Limitations:**

yes

**Quality:**

3

**Strengths And Weaknesses:**

Strengths: The problem is fundamental in machine learning theory. The paper is generally well-written. The problem investigated in the paper is clearly proposed and defined, the symbols and proofs are standard that easy to follow. The results are clear, with proper comparison and discussions to the results in related topics.

Weaknesses: I only have some suggestions:
  - Current draft seems somewhat "heavy" in math. The definitions of "different learning cases", "trichonomy rates" and notions of "at rate", "not faster than", all require large text space. Besides, the authors provide most of the proofs in main text. As a paper with page limit of 9, the main body of the paper is full of definitions, theorems and proofs, sometimes hard to follow. Is it possible to reduce the appearance of similar definitions or theorems, and put more proofs into the appendix?
  - While it is definitely OK to prove some of the results in this paper by adapting from [Hanneke and xu, 2024], it would be better if the authors could discuss how the proof techniques (rather than setting) in this paper differ from that of previous works.
  - The cross-marks in Fig.1 may incur some mis-understandings that the corresponding regions are empty, while the marked areas are proven to be non-empty. It would be better if they could be replaced by some other symbols.
  - The last section includes 8 paragraphs of discussions, while some of them are similar and some of them seems a bit minor. Fewer paragraphs of discussions with better organization of the topics may be better.

---

> ### Author Rebuttal · Authors · 2025-07-30
>
> Thank you for the feedback.
>
> > Current draft seems somewhat "heavy" in math. The definitions of "different learning cases", "trichonomy rates" and notions of "at rate", "not faster than", all require large text space. Besides, the authors provide most of the proofs in main text. As a paper with page limit of 9, the main body of the paper is full of definitions, theorems and proofs, sometimes hard to follow. Is it possible to reduce the appearance of similar definitions or theorems, and put more proofs into the appendix?
>
> We will add more exposition and explanation to the revised version.
>
>
> > While it is definitely OK to prove some of the results in this paper by adapting from [Hanneke and xu, 2024], it would be better if the authors could discuss how the proof techniques (rather than setting) in this paper differ from that of previous works.
>
> We thank the reviewer for prompting a more detailed comparison with the recent work of Hanneke and Xu (2024). While the combinatorial structures in our analysis resemble those used in their study of ERM, we stress that this connection is far from immediate. Conceptually, the two works address different questions: Hanneke and Xu analyze the performance of a specific algorithm—ERM— in the univsesal setting , whereas our work investigates *optimal learning rates* in a different, semi non-uniform context. The fact that related combinatorial dimensions appear in both analyses is surprising and not something that could have been anticipated a priori; uncovering this connection required significant effort.
>
> Even after identifying this link, our technical development diverges in important ways. Most notably, our fine-grained characterization of learning rates in terms of $\text{VCE}(H)/n$ introduces new analytical techniques. In particular, we make essential use of a novel application of König’s lemma to show that the version space eventually collapses to a class of VC dimension at most $\text{VCE}(H)$ on future instances. This argument plays a central role in our analysis and, to the best of our knowledge, does not appear in prior work.
>
> > The cross-marks in Fig.1 may incur some mis-understandings that the corresponding regions are empty, while the marked areas are proven to be non-empty. It would be better if they could be replaced by some other symbols.
>
> Thanks, we will improve Figure 1.
>
> > The last section includes 8 paragraphs of discussions, while some of them are similar and some of them seems a bit minor. Fewer paragraphs of discussions with better organization of the topics may be better.
>
> Thanks for the comment. We will improve the exposition here.
>
> > I am concerning the significance and applicability of theoretically studying the non-uniform settings of learning \emph{in the present days}. Specifically, casting at its greatest generality, what conclusions can be drawn from research in this direction, and what problems can be solved? How does resolving these issues promote the study of machine learning theory or assist in solving real-world machine learning problems? I appreciate that this is a fundamental research area that worth to investigate, but to be honest, such concern affect my score to some extent (and sadly, I can hardly tell a clear criteria of how this affects). I would appreciate it if the authors could share some opinions at this point.
>
> We believe that our results can provide guidance and improve our understanding when and how sample efficient learning from data is possible. We provide some theoretical support for the intuition that some distributions are easier to learn than others. In this sense our work can be placed into the broader theme of generalization guarantes **beyond the worst-case**. E.g., standard uniform PAC learning rates are typically “too worst-case” and in practice much better error rates are observed. Marginal non-uniform learning, similarly to universal learning and classic concept-nonuniform rates (e.g., structural risk minimization), can provide some understanding and explanation for these faster rates (and/or rates with much smaller constants) observed in real applications. This is possible as in such nonuniform/universal learning models a different order of quantifiers is used compared to standard PAC learnability, allowing it to be less focused on the worst-case. Overall this promotes the development of a generalization theory that is more flexible and expressive to capture properties of the data, which makes it easier to learn.
>
> > In line 106-107 you've mentioned that "there are families of classes such that $\operatorname{VCE}(\mathcal H)=1$ ", while $\operatorname{VC}(\mathcal H)$ is unbounded", while in line 135-136 you said that $1\le\operatorname{VCE}(\mathcal H)<\infty$ is equivalent to $|\mathcal H|=\infty$ and  $\operatorname{VC}(\mathcal H)<\infty$. Do these two statements mean that while there exists classes $\mathcal H$ with VCE-dimension 1 and arbitrarily large VC-dimension, there does not exist a class $\mathcal H$ with VCE-dimension $1$ and infinite VC-dimension?
>
> Exactly, $\operatorname{VCE}
> \leq \operatorname{VC}$ but for each $d \in\mathbb{N}$
> there is a class $\mathcal H_d$
> with $\operatorname{VC}(\mathcal
> H_d)=d$ and $\operatorname{VCE}
> (\mathcal H_d)=1$. However, $ \operatorname{VC}=\infty$ if and only if $\operatorname{VCE}=\infty$. Such a class $\mathcal H_d$ can be obtained by taking a disjoint union of shattered sets.

---

> > ### Comment · Reviewer_ycGG · 2025-08-05
> >
> > Thanks the authors for the detailed response. My concern has been properly resolved, and specifically, it convinced me that the paper fits within the borader theme of beyond-the-worst-case generalization analysis, which definitely serves as one of the necessary directions that push forward the study of learning theory. Therefore, I have updated the review and kept my score.

---

### Official Review · Reviewer_1Rvq · 2025-06-30

**Clarity:** 3
**Significance:** 2
**Originality:** 2
**Rating:** 4
**Confidence:** 4

**Summary:**

The paper explores one of the four fundamental notions of PAC learnability that has not been thoroughly studied: marginal-nonuniform learnability, where the generalization rate may depend on the marginal distribution but must be consistent across all concepts. The authors present a clear trichotomy of possible learning rates under this framework: 1) exponential rate for finite classes, 2) linear rates determined by the VC-eluder dimension, and 3) arbitrarily slow rates when the dimension is unbounded. The results are accompanied by multiple matching lower bounds, a detailed analysis of the linear case, and a contextualization within the broader landscape of universal learning. The paper is purely theoretical and offers useful characterizations of how distribution dependence affects learnability.

**Questions:**

Please take a moment to review the comments and suggestions in the sections above. Let me know if anything seems missing, misunderstood, or could benefit from further clarification. Additionally,
- Could the authors elaborate on whether the results are applicable in agnostic/non-realizable settings?
- What common assumptions could cause the VC-eluder dimension to be significantly smaller than the VC dimension?
- Are there specific examples that match the theoretical findings to illustrate the idea that "some data distributions are inherently easier to learn from than others"?

**Ethical Concerns:**

["NO or VERY MINOR ethics concerns only"]

**Final Justification:**

I assigned a score of 4 and supported the paper's acceptance because it presents "a precise trichotomy, providing a comprehensive classification of learning rates based on a single combinatorial parameter, the VC-euler dimension."

To receive more substantial support, the authors should demonstrate novelty in some ways, such as in formulation or technical strategies. I understand that researchers often build their results on top of prior works and ideas. Still, a relatively straightforward extension or application of existing theorems and tools does not meet the novelty bar.

That being said, I thank the authors again for the helpful discussion and promised revisions, including adding more insights regarding the communicated points.

**Limitations:**

This work is purely theoretical and has minimal negative impact on society. The authors have already identified some gaps in their results and future directions, such as active learning settings, which should help address the limitations.

**Paper Formatting Concerns:**

The paper is well-organized and consistently formatted, with notation maintained throughout. Including a section on technical novelty would be beneficial, but that's mainly related to the content.

**Quality:**

3

**Strengths And Weaknesses:**

Strengths:
- The paper examines a natural variation of nonuniform PAC learning that is both conceptually interesting and technically well-defined.
- The main accomplishment is a precise trichotomy, providing a comprehensive classification of learning rates based on a single combinatorial parameter, the VC-euler dimension.
- The presentation is clear, especially in the abstract and introduction, and the proofs use existing tools from prior work transparently.
- The paper connects to established frameworks in universal and concept-non-uniform learning, effectively placing its contribution within the existing literature.

Weaknesses:
- The formulation of marginal-nonuniform learning is not entirely new. It originates from the taxonomy introduced by Ben-David et al. (1995), and the paper does not distinguish itself in terms of foundational definitions.
- Many of the theoretical claims, including several lemmas and various example-based separations, directly build on existing results in the literature, which seem to require only modest adjustments rather than significant innovation.
- The fine-grained analysis of the linear rate regime using the VC-eluder dimension appears to be the most technically challenging part of the paper (e.g., Lemmas 10-12). It would be helpful if the authors more clearly highlighted where genuine technical novelty appears.

---

> ### Author Rebuttal · Authors · 2025-07-30
>
> Thanks for your feedback.
>
> > The formulation of marginal-nonuniform learning is not entirely new. It originates from the taxonomy introduced by Ben-David et al. (1995), and the paper does not distinguish itself in terms of foundational definitions.
>
> We will emphasize the distinction to Ben-David et al. (1995) more clearly in the revised version. Ben-David et al. introduced the marginal nonuniform setting; we characterized all possible rates and when they can occur. The fact that the model was studied before makes our results arguably more interesting in our opinion.
>
> > Many of the theoretical claims, including several lemmas and various example-based separations, directly build on existing results in the literature, which seem to require only modest adjustments rather than significant innovation.
> The fine-grained analysis of the linear rate regime using the VC-eluder dimension appears to be the most technically challenging part of the paper (e.g., Lemmas 10-12). It would be helpful if the authors more clearly highlighted where genuine technical novelty appears.
>
> We thank the reviewer for this thoughtful remark and for highlighting the fine-grained analysis of the linear rate regime as a technically challenging contribution.
>
> From a technical standpoint, the closest prior work is that of Hanneke and Xu (2024), who study the performance of ERM in the universal learning setting. While some of the combinatorial structures that appear in both works are related, the connection is far from immediate. Their work focuses on the behavior of a specific algorithm—ERM—whereas our work addresses the more fundamental problem of characterizing optimal learning rates in an online, non-realizable setting. The emergence of similar combinatorial dimensions in these distinct contexts was not something we anticipated, and uncovering this link required significant conceptual effort.
>
> Even after this connection was identified, the technical approaches differ substantially. A key novelty in our work is the precise characterization of the linear rate regime via the VC-eluder dimension. In particular, Lemmas 10–12 hinge on a new and central argument based on König’s lemma, which we use to show that the version space eventually becomes confined to a subclass of VC dimension at most $\mathrm{VCE}(H)$. To our knowledge, this application of König’s lemma is new in the learning theory literature and plays a critical role in our analysis. We are pleased that the reviewer found this component to be technically challenging—it is indeed one of the central contributions of the paper.
>
>
> > Could the authors elaborate on whether the results are applicable in agnostic/non-realizable settings?
>
> No our approach does not immediately apply in the agnostic setting. We believe that this would likely require considerable effort and novel techniques. Nevertheless, we think that the realizable case is already technically rich and substantial by itself.
>
> The agnostic case is of course interesting. There is even an interesting question about what is the appropriate formulation of the setting for the agnostic case. For instance, we could consider the supremum over all conditional distributions, or we could consider only deterministic labels and consider the supremum over all labeling functions. This is an interesting direction for future work.
>
> > What common assumptions could cause the VC-eluder dimension to be significantly smaller than the VC dimension?
>
> Great question! A simple necessary condition for the VC-eluder dimension to be large is an infinite number of large disjoint shatterable sets. Thus, if the VC dimension is large, however only a bounded number of large disjoint sets are shatterable, the VC-eluder dimension will be small (or even 0). For example finite classes can have $\operatorname{VC}$ up to $\log|\mathcal{H}|$ while $\operatorname{VCE}=0$ always.
>
> > Are there specific examples that match the theoretical findings to illustrate the idea that "some data distributions are inherently easier to learn from than others"?
>
> The proof of our $\operatorname{VCE}(\mathcal H)/n$ upper bound might provide some intuition. The marginal-dependent constants are mostly given by certain tail probabilities indicating when all version spaces will have bounded VC dimension on the data sequence with constant probability. In some sense samples from worst-case distributions behave in a way such that the version space (for a specific labeling) continues to have large VC dimension. By contrast, "simpler" distributions will produce samples that likely result in small VC dimension of the version space on data sets which can be sampled from the distribution.
>
> As a simple example of a distribution where the dimension of learning is smaller than the worst-case distribution (even before restricting to a version space), consider the class of linear classifiers in $\mathbb{R}^d$ but with a distribution supported on a $k < d$ dimensional subspace.

---

> > ### Comment · Area_Chair_zzEh · 2025-08-05
> >
> > Dear reviewer 1Rvq,
> >
> > The authors have provided detailed responses to the points you mentioned. Please review the author's response to check if the concerns you mentioned are addressed, and follow up as necessary so that authors have a chance to respond. Please do not wait till the last minute, and also do not forget to acknowledge that you have read the reviews. Thanks!
> >
> > -AC

---

> > ### Comment · Reviewer_1Rvq · 2025-08-07
> >
> > Thanks for the response.
> > > Review - The formulation of marginal-nonuniform learning is not entirely new. It originates from the taxonomy introduced by Ben-David et al. (1995), and the paper does not distinguish itself in terms of foundational definitions.
> >
> > > Rebuttal - Ben-David et al. introduced the marginal nonuniform setting ... The fact that the model was studied before makes our results arguably more interesting.
> >
> > The original review point mainly was about the novelty of the formulation. I acknowledge the theoretical contributions made by the authors; however, this doesn't address my concerns.
> >
> > > Review - Many of the theoretical claims, including several lemmas and various example-based separations, directly build on existing results in the literature, which seem to require only modest adjustments rather than significant innovation. The fine-grained analysis of the linear rate regime using the VC-eluder dimension appears to be the most technically challenging part of the paper (e.g., Lemmas 10-12). It would be helpful if the authors more clearly highlighted where genuine technical novelty appears.
> >
> > > Rebuttal - The closest prior work is that of Hanneke and Xu (2024) ... while some of the combinatorial structures that appear in both works are related, the connection is far from immediate.
> >
> > > Rebuttal - A key novelty in our work is the precise characterization of the linear rate regime via the VC-eluder dimension.
> >
> > I think neither of the points addresses the review's concerns regarding technical novelty, which primarily involves the introduction of new technical tools, proof strategies, or technical connections. For example, the second rebuttal point focused mainly on the contributions/results rather than the techniques.
> >
> > While "Lemmas 10–12 hinge on a new and central argument based on König's lemma" does fall into this domain, it is limited to a relatively narrow portion of the proof.
> >
> > > Review questions:
> > > 1. Could the authors elaborate on whether the results are applicable in agnostic/non-realizable settings?
> > > 2. What common assumptions could cause the VC-eluder dimension to be significantly smaller than the VC dimension?
> > > 3. Are there specific examples that match the theoretical findings to illustrate the idea that "some data distributions are inherently easier to learn from than others"?
> >
> > Thanks for providing detailed responses to the questions. It would be great if the authors could incorporate some of the insights from their comments into the main paper. Thanks!

---

> > > ### Author Response · Authors · 2025-08-08
> > >
> > > Thanks for the follow up. We will definitely add the insights from the rebuttal to the revised version.
> > >
> > > On technical novelty (besides the fine-grained results): Our main technical challenge in, say, our lower bound proofs was to identify a single marginal distribution that works for all $n$ simultaneously. Here the explicit connection between infinite (VC)-eluder sequences and learning with fixed marginal distributions could be considered a technical novelty. This is different to standard uniform PAC/no free lunch lower bounds, where for each $n$ a different distribution can be used. While our techniques happen to be similar to previous approaches, e.g., to Hanneke and Xu (2024) and other PAC lower bounds, our wording in "adapted from Lemma XY from Hanneke and Xu (2024)" was probably somewhat misleading. It is not the case that our results follow immediately from theirs (for example, our lower bounds hold for arbitrary learners, while theirs only for specific worst-case ERMs). We will clarify the phrasing here in the revised version.
> > >
> > > In general we believe that most theoretical research is largely based on identifying the right previous ideas and techniques to then carefully adapt and combine them in insightful ways. It is quite rare that truly novel proof techniques are discovered.
> > >
> > > We hope we addressed your remaining concerns and thank the reviewer again for the thorough evaluation.

---

### Official Review · Reviewer_f8zt · 2025-07-03

**Clarity:** 4
**Significance:** 3
**Originality:** 3
**Rating:** 4
**Confidence:** 4

**Summary:**

This paper formalizes "marginal-nonuniform learning" as a variant of the PAC model that allows for distribution-specific learning rates that hold uniformly over all possible target concepts. Specifically, a concept class is learnable at rate R if for every marginal distribution over examples, there exists a (distribution-dependent) constant c such that every concept in the class is learnable with error at most ~R(cn) using n samples. This ordering of quantifiers makes it a relaxation of distribution-free PAC learning (uniform), a strengthening of universal learning, and complementary to (concept)-nonuniform learning.

The paper proves the following results. The main result is a trichotomy of rates for marginal-nonuniform learning in terms of the VC-eluder (VCE) dimension of the target class. Moreover, in the case where the learning rate is linear, it scales as VCE(H)/n. The paper then revisits concept-nonuniform learning, for which it shows that the one-inclusion graph algorithm achieves a linear rate for countable unions of VC classes. Finally, the paper discusses the relationships between uniform, non-uniform, and universal learning and presents examples of classes separating these notions.

**Questions:**

* Does Theorem 16 (or its proof) suggest a characterization of the fine-grained rate of concept-nonuniform learnability of countable unions of VC classes?

**Ethical Concerns:**

["NO or VERY MINOR ethics concerns only"]

**Final Justification:**

This is a nice conceptual submission which, based on discussion with the authors, I am confident they can refine into a strong presentation.

**Limitations:**

Yes.

**Paper Formatting Concerns:**

None.

**Quality:**

4

**Strengths And Weaknesses:**

(+) The paper studies a mathematically natural learning model that cleanly fills in a gap between previously studied ones.

(+) The main trichotomy result and characterization via VCE dimension is sharp and satisfying.

(+) Proofs are conceptually simple and do a nice job intuitively illuminating why the corresponding statements are true.

(-) While the question of understanding learning rates for marginal-nonuniform learners is new, it's debatable how new the model of marginal-nonuniform learning itself is. As acknowledged in the paper, it's related to distribution-specific and distribution-family learning. But even more closely related questions about the difference between uniform PAC learning and learnability with respect to known, but arbitrary, marginal distributions have been studied in the past. For example, in Section 7.2 of the arXiv version of "A General Characterization of the Statistical Query Complexity," Feldman showed a strong separation between distribution-independent SQ complexity and the max over all distributions of distribution-specific SQ complexity.

(-) The paper's statements and proofs very closely track those of recent prior work of Hanneke and Xu on universal learning rates. As such, there isn't much in the way of new technical ideas or tools developed in this paper.

---

> ### Author Rebuttal · Authors · 2025-07-30
>
> Thank you for your feedback.
>
> > While the question of understanding learning rates for marginal-nonuniform learners is new, it's debatable how new the model of marginal-nonuniform learning itself is. As acknowledged in the paper, it's related to distribution-specific and distribution-family learning. But even more closely related questions about the difference between uniform PAC learning and learnability with respect to known, but arbitrary, marginal distributions have been studied in the past. For example, in Section 7.2 of the arXiv version of "A General Characterization of the Statistical Query Complexity," Feldman showed a strong separation between distribution-independent SQ complexity and the max over all distributions of distribution-specific SQ complexity.
>
> Thanks for bringing Feldman’s results to our attention. We will add a more thorough  discussion of related work to the revised version. We believe that the fact that this model was studied before makes our result arguably more interesting.
> Our main starting point was the taxonomy of Ben-David et al. (1995) (on different nonuniform learnability models). We believe that establishing possible error rates for the settings proposed there is valuable.
>
> > The paper's statements and proofs very closely track those of recent prior work of Hanneke and Xu on universal learning rates. As such, there isn't much in the way of new technical ideas or tools developed in this paper.
>
> While the combinatorial structures used in our work resemble those in the analysis of ERM by Hanneke and Xu (2024), we emphasize that this connection is far from obvious. The two settings are conceptually quite distinct: ERM concerns a specific learning algorithm, while our work addresses the more fundamental question of optimal learning rates in a different, non-uniform setting. That similar combinatorial dimensions emerge in both contexts is not something that could have been foreseen—it took us considerable effort to uncover this link.
>
> Furthermore, even once this connection was recognized, the technical paths diverge significantly. A particularly notable difference lies in our fine-grained analysis of learning rates in terms of $\text{VCE}(H)/n$, which introduces new ideas—most prominently, the use of König’s lemma to show that the version space will eventually have VC dimension at most $\text{VCE}(H)$ on future instances. This use of König’s lemma is novel and central to our analysis.
>
> > Does Theorem 16 (or its proof) suggest a characterization of the fine-grained rate of concept-nonuniform learnability of countable unions of VC classes?
>
> Good question! Let us first unpack the interpretation of the question.  In the standard sense of fine-grained rates (Bousquet et al., 2023), a fine-grained rate of, say, linear $1/n$ are expressed as a bound $d/n$ holding for all sufficiently large $n$ (dependent on the distribution and target concept), where $d$ is only dependent on the class $\mathcal{H}$ and not the distribution.  In the concept-nonuniform case the achieved rate is $\operatorname{VC}(\mathcal H_f)/n$ where $\mathcal H_f$ is the “smallest” class containing the target concept $f$.  In some sense, this is a kind of fine-grained rate, but there is perhaps a more-natural analogue to the sense studied by Bousquet et al., 2023: Namely, to express a fine-grained concept-nonuniform rate, we would require a bound $d/n$ to hold for all sufficiently large $n$, where "sufficiently large" may only depend on $f$ (not the marginal on $x$) and where $d$ may only depend on the class $\mathcal{H}$.  For this definition, we can argue that the smallest such $d$ possible is proportional to the VC dimension of $\mathcal{H}$ (or else such a constant $d$ does not exist if the VC dimension is infinite).  To see this, consider any shattered set of some size $k$, and for each sample size $n$ consider a marginal distribution which puts $1-O(k/n)$ mass on one point and $O(1/n)$ mass on each of the remaining $k-1$ points.  By standard lower bound arguments, for any learning algorithm, among the $2^{k-1}$ concepts which shatter these $k-1$ points while labeling the other point $0$, there exists one for which the expected error rate is $\Omega(k/n)$. By the pigeonhole principle, among these $2^{k-1}$ functions, there exists at least one function witnessing this $\Omega(k/n)$ lower bound for infinitely many $n$ (so that the marginal distribution in these lower bounds varies with $n$ but the target concept does not).  Therefore, for this concept, any $d/n$ lower bound holding for all sufficiently large $n$ must have $d = \Omega(k)$, and since $k$ is simply the size of a shattered set, the conclusion follows.

---

> > ### Comment · Reviewer_f8zt · 2025-08-05
> >
> > Thank you for the responses, which address all of my questions and confirm my generally positive impression of the paper.

---

### Official Review · Reviewer_SdTA · 2025-07-24

**Clarity:** 3
**Significance:** 3
**Originality:** 3
**Rating:** 5
**Confidence:** 4

**Summary:**

This paper discusses the question marginal-nonuniform learnability, requiring learners to depend only non-uniformly on the marginal, but uniformly on the concept. Thus this notion contrast standard non-uniform learning, which requires guarantees to be uniform for all marginals but non-uniform over concepts. These notions are in-between notion of uniform PAC learning, which provides uniform rates for all concepts and distributions and universal learning, which allow for non-uniform guarantees for both concept and marginal.

They relate this learnability to a recently introduced complexity measure - the VC-eluder dimension. In particular they show that:
- A class is marginal non-uniformly learnable, with exponential rates if and only if it is finite
- A class is marginal non-uniformly learnable with linear rates if and only if the class has finite VC-eluder dimension
- A class has at best arbitrarily slow non-uniform rates for marginal non-uniform learnability if it has infinite VC-eluder dimension.

They also show improved rates for concept-nonuniform learning.

Their results rely on established methods within the literature of universal learning rates.

**Questions:**

One downside of non-uniform learning approaches as opposed to uniform PAC learning approaches is that non-uniform learning guarantees do not provide any error/confidence guarantees for a learned classifier.
Is this something you can still overcome in the marginal non-uniform setting?

Can you say anything about learnability in the agnostic case?

**Ethical Concerns:**

["NO or VERY MINOR ethics concerns only"]

**Limitations:**

The authors provide proofs for all of their statements. Limitations and assumptions have been acknowledged in the main text of the paper.

**Paper Formatting Concerns:**

No concerns.

**Quality:**

3

**Strengths And Weaknesses:**

Strengths:

I overall enjoyed the presentation of the paper. I enjoyed the use of color to highlight different quantifiers as well as the picture separating between different cases.
There is a good overview of the literature, highlighting many important results.
The results discuss a relevant non-uniform learnability notion and it provides novel results.
I enjoyed the discussion section.

Overall I enjoyed reading the paper and found it insightful and well-rounded.


Weaknesses:
-In contrast to the universal learning rates, this work does not provide a full characterization, as there is no distinction between arbitrarily slow rates and a class not being non-uniform marginally learnable. In universal learning every class (with sufficient measurability conditions) is universally learnable at least with arbitrarily slow rates. As the authors mention, in previous work it has been shown that this is not the case for marginal non-uniform learning. However there is no clear characterization of what distinguishes marginal non-uniformly learnable classes from not marginal non-uniformly learnable classes. I think this should be highlighted more, as the presentation as trichotomy which is akin to the universal learning rates paper from Bousquet et al 2021 (and follow-up works), where the trichotomy gives a more complete characterization (as there is no fourth case of unlearnable classes)
- I wish there was a bit more intuition given for cases in which marginal non-uniform learnability is distinct from universal learning. There are some examples mentioned from the literature, but it would be nice if they could be discussed in more detail. How do the complexity measures infinite Littlestonetree/VCL-tree and VC-eluder dimension relate? A lot of the examples from old work are merely stated, which makes sense for space constraints. In order to understand the intuition of this paper, it might help though to rewrite those examples and relate them explicitly to those complexity measures.
- The insight of this paper heavily relies on bringing together insights from previous work and the ideas within this work itself aren't as novel or ground-breaking. (I don't view this as a reason to reject the paper, as it was still insightful to read the paper)
- As is often the case for results on universal learning rates the results rely on the realizable case, which is arguably a very strong assumption. (However, as this is a general drawback of this line of work I would not count it as heavily against this work in particular)


Overall I feel that this was a decent paper and I enjoyed reading it. The characterization and discussion is a good addition to the literature and fits well within the NeurIPS program.

---

> ### Author Rebuttal · Authors · 2025-07-30
>
> Thanks a lot for your encouraging feedback!
>
> > In contrast to the universal learning rates, this work does not provide a full characterization, as there is no distinction between arbitrarily slow rates and a class not being non-uniform marginally learnable.
>
> We agree that the learnability question is interesting and view it as an intriguing direction for future work.
> Nevertheless, our characterization is complete in the case of countable $X$, where learnability always holds. For uncountable $X$, the outcome depends on the choice of sigma-algebra, and some measure-theoretic distinctions are required. For instance, under the power set sigma-algebra, learnability is always possible, as only discrete distributions are allowed.
> Moreover, notice that our results do provide a complete characterization of what optimal learning rates are possible—exponential, linear, arbitrarily slow, and non-learnable.
>
> > I wish there was a bit more intuition given for cases in which marginal non-uniform learnability is distinct from universal learning. There are some examples mentioned from the literature, but it would be nice if they could be discussed in more detail. How do the complexity measures infinite Littlestonetree/VCL-tree and VC-eluder dimension relate?
>
> We thank the reviewer for this insightful question—we agree that elaborating on the distinctions between marginal non-uniform and universal learning would add clarity, and we will include a more detailed discussion in the next revision.
>
> There are some clean implications in both directions. On one hand, any class that admits a linear rate in the marginal non-uniform setting must have finite VC dimension, and therefore admits at most a linear rate in the universal setting. Similarly, any class with an exponential rate in the marginal non-uniform setting must be finite, and hence also enjoys an exponential rate in the universal setting.
>
> On the other hand, there exist classes that achieve exponential rates in the universal setting but are arbitrarily slow in the marginal non-uniform setting. This occurs, for example, when a class has no infinite Littlestone tree but has infinite VC dimension. Such examples are discussed in the work of Bousquet et al.; one nice such example is that of monotone functions over the grid of natural numbers $\mathbb{N}^d$, and wrt the natural partial order over grid points.
>
> We will expand on these points and add more such concrete examples to our examples section in the revised version.
>
> > The insight of this paper heavily relies on bringing together insights from previous work and the ideas within this work itself aren't as novel or ground-breaking (I don't view this as a reason to reject the paper, as it was still insightful to read the paper)
>
> While the combinatorial structures used in our work resemble those in the analysis of ERM by Hanneke and Xu (2024), we emphasize that this connection is far from obvious. The two settings are conceptually quite distinct: ERM concerns a specific learning algorithm, while our work addresses the more fundamental question of optimal learning rates in a different, non-uniform setting. That similar combinatorial dimensions emerge in both contexts is not something that could have been foreseen—it took us considerable effort to uncover this link.
>
> Furthermore, even once this connection was recognized, the technical paths diverge significantly. A particularly notable difference lies in our fine-grained analysis of learning rates in terms of $\text{VCE}(H)/n$, which introduces new ideas—most prominently, the use of König’s lemma to show that the version space will eventually have VC dimension at most $\text{VCE}(H)$ on future instances. This use of König’s lemma is novel and central to our analysis.
>
> > As is often the case for results on universal learning rates the results rely on the realizable case, which is arguably a very strong assumption. (However, as this is a general drawback of this line of work I would not count it as heavily against this work in particular)
>
> The agnostic case is of course interesting.  There is even an interesting question about what is the appropriate formulation of the setting for the agnostic case.  For instance, we could consider the supremum over all conditional distributions $Y|X$, or we could consider only deterministic labels and consider the supremum over all labeling functions.  This is an interesting direction for future work.
>
> > One downside of non-uniform learning approaches as opposed to uniform PAC learning approaches is that non-uniform learning guarantees do not provide any error/confidence guarantees for a learned classifier. Is this something you can still overcome in the marginal non-uniform setting?
>
> We agree. Note that marginal-nonuniform learning also provides bounds in the case when the marginal distribution is known (e.g., uniform over sphere, etc.) or comes from a known family of distributions. In such cases, at least in principle, the marginal-dependent constants in our bounds could be computable/boundable (they are mostly given by certain tail probabilities indicating when all version spaces will have bounded VC dimension with constant probability).
>
> > Can you say anything about learnability in the agnostic case?
>
> See above.

---

### Decision · Program_Chairs · 2025-09-17

**Decision:**

Accept (poster)

**Comment:**

This paper considers the question of marginal-nonuniform learnability, requiring learners to depend uniformly on the target concept but only non-uniformly on the marginal distribution.

The reviewers found the paper to be clearly written and the results, which fill in the gap between previously studied models, to be sharp and satisfying. Hence, this would be a nice addition to NeurIPS.

The reviewers had several suggestions, which I expect the authors to address in the camera-ready version. Here are a few of these suggestions, but I encourage the reviewers to apply as many of these suggestions as possible. (1) The proofs closely follow prior work (Hanneke and Xu 2024), so it would be good to highlight the difference more clearly. (2) Please cite and discuss Feldman's work, which is missing. (3) Please specify the origin of the model studied in this paper more clearly. (4)  Please provide a bit more intuition or cases in which marginal non-uniform learnability is distinct from universal learning.